# MIAM: Modality Imbalance-Aware Masking for Multimodal Ecological Applications

**Robin Zbinden,**\* **Wesley Monteith-Finas,**\* **Gencer Sumbul, Nina van Tiel,**
**Chiara Vanalli, Devis Tuia**
École Polytechnique Fédérale de Lausanne (EPFL), Switzerland

## Abstract

Multimodal learning is crucial for ecological applications, which rely on hetero-geneous data sources (e.g., satellite imagery, environmental time series, tabular predictors, bioacoustics) but often suffer from incomplete data across and within modalities (e.g., unavailable satellite image due to cloud cover, missing records in a time series). While data masking strategies have been used to improve robust-ness to missing data by exposing models to varying input subsets during training, existing approaches typically rely on static masking and inadequately explore the space of input combinations. As a result, they fail to address modality imbalance, a critical challenge in multimodal learning where dominant modalities hinder the optimization of others. To fill this gap, we introduce Modality Imbalance-Aware Masking (MIAM), a dynamic masking strategy that: (i) explores the full space of input combinations; (ii) prioritizes informative or challenging subsets; and (iii) adaptively increases the masking probability of dominant modalities based on their relative performance and learning dynamics. We evaluate MIAM on two key ecological datasets, GeoPlant and TaxaBench, with diverse modality config-urations, and show that MIAM significantly improves robustness and predictive performance over previous masking strategies. In addition, MIAM supports fine-grained contribution analysis across and within modalities, revealing which vari-ables, time segments, or image regions most strongly drive performance.

## 1 Introduction

Ecological modeling plays a central role in conservation, climate change adaptation, and environ-mental management (Pollock et al., 2025). Capturing complex ecological processes requires data that reflect multiple facets of both the environment and the species of interest. Consequently, eco-logical datasets are inherently multimodal (Hartig et al., 2024; Miao et al., 2025), integrating diverse inputs such as tabular environmental variables (e.g., elevation, soil properties), time series (e.g., cli-mate records), audio (e.g., bioacoustics), natural images (e.g., species observations), and satellite imagery (Picek et al., 2024; Sastry et al., 2025). Learning effectively from this heterogeneous data presents several challenges. First, ecological data are frequently incomplete due to limitations in data collection, such as cloud-obstructed satellite images or temporal gaps in monitoring efforts. Missing data can occur at the modality level (e.g., no image available for a location) or within modalities (e.g., missing entries in a climate time series). Second, quantifying the importance of different in-puts is critical, as these models also aim to provide ecological insights. This includes contributions both across modalities (e.g., how useful is satellite imagery vs. tabular data?) and within them (e.g., which year in a time series matters the most?). Addressing these challenges requires models that can flexibly operate on arbitrary and incomplete subsets of inputs.

Recent advances in multimodal learning have made progress toward this goal through data masking, where a *masking strategy* specifies a probability distribution over which inputs are hidden from the model during training. Models such as 4M (Mizrahi et al., 2023; Bachmann et al., 2024) implement this idea by randomly masking subsets of inputs, simulating missing data. This exposes the model to diverse modalities and feature combinations, promoting robustness to incomplete inputs and en-abling contribution techniques such as Shapley-based feature importance (Zbinden et al., 2026).

---

\*Equal contribution. Correspondence to: `robin.zbinden@epfl.ch`

However, these masking distributions do not adequately explore the space of input subsets and, being typically fixed and uniform, do not adapt to evolving learning dynamics or modality-specific characteristics during training.

As a result, such approaches do not address a crucial challenge in multimodal learning: *modality imbalance* (also known as *modality competition*). This occurs when some modalities dominate the learning process, capturing most of the predictive signal and gradient flow, thereby impeding the optimization of other, potentially complementary, modalities (Wang et al., 2020; Huang et al., 2022; Wu et al., 2022). An example of modality imbalance is shown in Fig. 1. Several mitigation strategies have been proposed, including gradient reweighting (Peng et al., 2022), knowledge distillation from unimodal teachers (Du et al., 2021), and adaptive training schedules based on per-modality learning speeds (Wu et al., 2022). However, these methods often require additional components or supervision, while even simple modality dropout (Neverova et al., 2015) has proven to be competitive with more complex approaches. Building on this, Wei et al. (2024) proposed On-the-fly Prediction Modulation (OPM), a masking strategy that adjusts per-modality probabilities based on relative performance scores. However, these scores remain nearly static during training, operate only at the modality level (masking an entire modality or none of it), and fail to fully explore the space of possible input combinations.

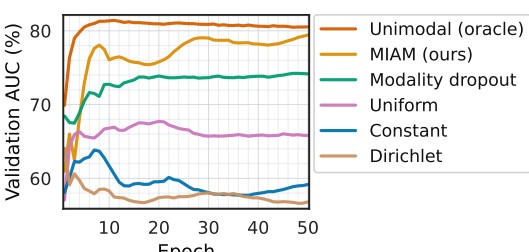

Figure 1: **Modality imbalance.** On the GeoPlant dataset, multimodal masking-based approaches that ignore modality imbalance underperform a unimodal model when evaluated using satellite imagery only, as dominant modalities hinder effective optimization. MIAM closes this gap by adaptively increasing the masking probability of dominant modalities based on their relative performance and learning dynamics.

In this work, we propose Multimodal Imbalance-Aware Masking (MIAM), a principled, dynamic, score-driven masking strategy illustrated in Fig. 2. We first formalize masking strategies as probability distributions over unit hypercubes and identify three key properties often missing from existing approaches. An effective masking strategy should have full support on the hypercube, prioritize corners while assigning higher weight to those corresponding to key input configurations, and adapt to modality imbalance by adjusting masking probabilities based on modality dominance. These insights motivate MIAM, which is designed to: i) handle arbitrary missing inputs; ii) mitigate modality imbalance by adjusting masking of dominant modalities; and iii) support both within and across modality contributions. To achieve this, MIAM constructs a mixture of product beta distributions to define masking probabilities and dynamically adjusts this distribution during training based on modality-specific performance and learning speed. We evaluate MIAM on two ecological benchmarks: GeoPlant (Picek et al., 2024) for species distribution modeling and TaxaBench (Sastry et al., 2025) for multimodal species classification, spanning three and five modalities, respectively. MIAM consistently outperforms existing masking approaches, with particularly strong gains for modalities affected by modality imbalance. Beyond predictive performance, MIAM also provides ecological insight by revealing not only which modalities are the most influential, but also which predictors, temporal segments, or image regions drive model performance – highlighting key ecological signals such as NDVI and heatwaves. Together, these results underscore the importance of principled masking in multimodal learning, and particularly so in ecological applications where data are heterogeneous and incomplete. Code available at `https://github.com/zbirobin/MIAM`.

## 2 RELATED WORK

**Multimodal learning** aims at integrating information from heterogeneous data sources such as audio, image, video, text, and tabular data (Baltrušaitis et al., 2018; Uppal et al., 2022; Xu et al., 2023; Zhang et al., 2023; Bachmann et al., 2024; Zong et al., 2025). Ideally, different modalities provide synergistic views of the underlying process (Dufumier et al., 2025), as in ecology, where multimodal approaches are increasingly adopted to leverage the diversity of available data sources (Miao et al., 2025; Hartig et al., 2024; Picek et al., 2024). However, combining multiple modalities often in-

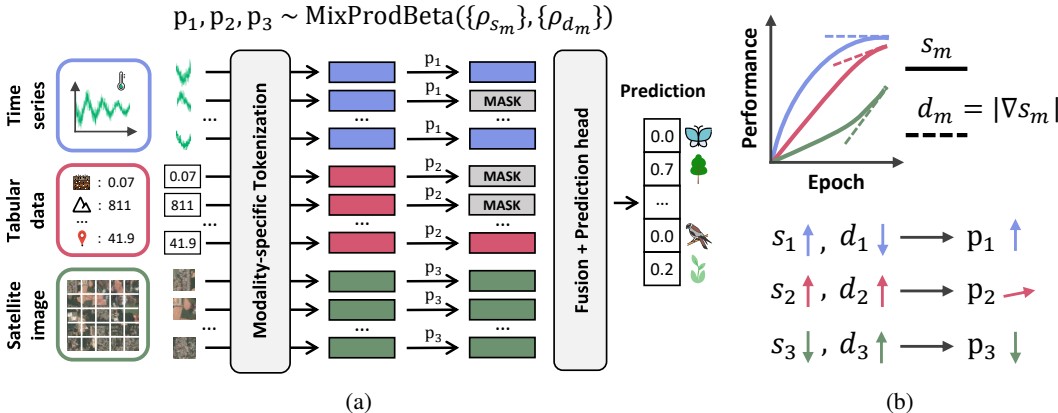

Figure 2: **Overview of MIAM.** (a) Each token of modality $m$ is masked with probability $\mathbf{p}_m$, sampled from a mixture of product beta distributions. (b) The distribution parameters are modulated by $\rho_{s_m}$ and $\rho_{d_m}$, derived from the per-modality performance $s_m$ and its absolute derivative $d_m$ to address modality imbalance. Modalities with relatively high $s_m$ and low $d_m$ are masked more often.

troduces the challenge of *modality imbalance*, where dominant modalities impede the optimization of less informative ones. This issue can arise from differences in predictive strength, input scale, or learning speed (Wang et al., 2020; Peng et al., 2022; Huang et al., 2022). Wang et al. (2020) shows that multimodal models can underperform their unimodal counterparts due to differing generalization rates across modalities, and proposed gradient blending to address this effect. Building on this, Peng et al. (2022) introduces On-the-fly Gradient Modulation (OGM) to adjust gradients based on the discrepancy of modalities' contributions, while Wu et al. (2022) proposed adapting training schedules based on modality-specific learning speeds to counteract greedy optimization behaviors. Other approaches adjust training using unimodal teacher distillation Du et al. (2021) or prototype-based regularization (Fan et al., 2023). While effective when all modalities are present, these methods assume complete inputs and are not designed to cope with missing modalities or arbitrary subsets – conditions that are especially common in ecological datasets.

**Masking** is widely used in self-supervised learning (SSL) as a pretext task to learn robust, general-purpose representations. By reconstructing masked inputs from partial context, models can learn without labels and develop deeper contextual understanding (Devlin et al., 2019; He et al., 2022). Beyond SSL, masking also promotes robustness to missing inputs. In supervised settings, modality dropout (Neverova et al., 2015) – randomly masking modalities during training – has shown competitive performance for handling missing modalities. MultiMAE (Bachmann et al., 2022) and 4M (Mizrahi et al., 2023; Bachmann et al., 2024) extend this idea by masking and reconstructing both across and within modalities, enabling models to flexibly handle arbitrary subsets of inputs while also supporting per-modality performance analysis. Similarly, Covert et al. (2023) leverages masking to estimate image patch contributions through Shapley values. In ecology, MaskSDM (Zbinden et al., 2026) applies uniform random masking to tabular predictors and satellite embeddings, improving robustness to missing data both across and within modalities. In another direction, Wei et al. (2024) adapts masking to address modality imbalance by adjusting dropout probabilities based on per-modality performance scores. Despite their success, most of the existing masking strategies rely on uniform distributions over a limited subset of input combinations, overlooking modality imbalance and lacking robustness to arbitrary missing inputs. In Section 3.1, we formalize existing masking strategies, identify their limitations, and propose key principles for effective masking in multimodal settings that serve as the foundation for MIAM.

## 3 METHODOLOGY

### 3.1 PROBLEM SETUP AND MASKING FORMULATION

In our multimodal setup, each input sample $x$ consists of $M$ modalities, represented as a tuple $x = (x^1, x^2, \ldots, x^M)$, where each modality $x^m$ is associated with $T_m$ tokens: $x^m =$

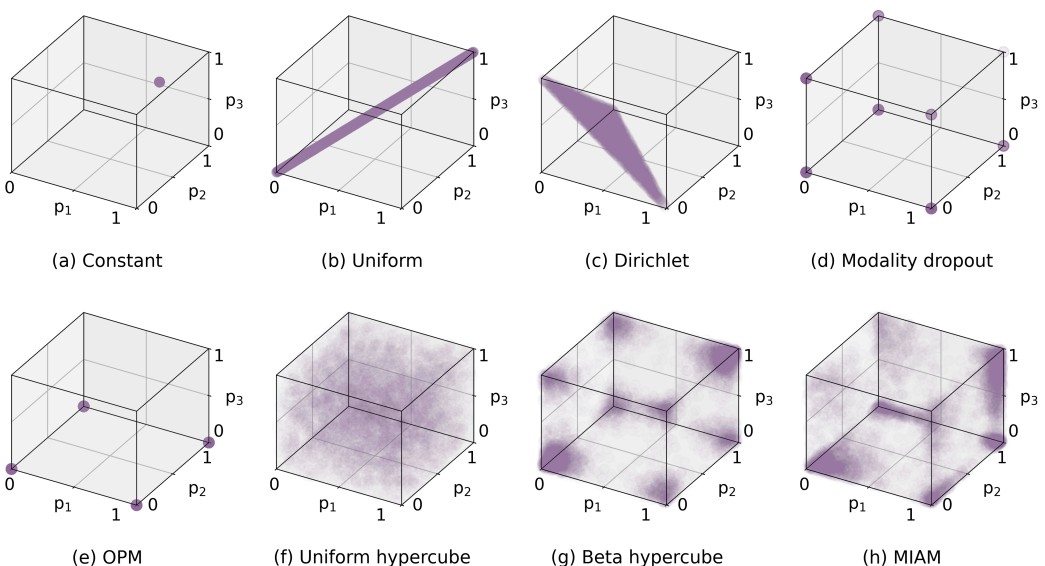

Figure 3: **Masking strategies** viewed as probability distributions $\mathbf{p} = (p_1, p_2, p_3)$ over the unit hypercube, illustrated with three modalities where modality 3 is dominated. $5000$ points are drawn per strategy to visualize the distribution. **(a)** Constant masking (e.g., 0.75) **(b)** Shared probability $p \sim \mathcal{U}(0, 1)$ across modalities (Zbinden et al., 2026) **(c)** Symmetric Dirichlet with $\boldsymbol{\alpha} = \mathbf{1}_M$ (Mizrahi et al., 2023). **(d)** Modality dropout with fixed probability (Neverova et al., 2015) **(e)** OPM (dynamic masking): modality 3 is never masked (Wei et al., 2024) **(f)** Uniform hypercube: independent $p_m \sim \mathcal{U}(0, 1)$ per modality for full support. **(g)** Beta hypercube: mixture of product beta distributions to prioritize corners **(h)** MIAM (dynamic masking): beta hypercube with imbalance-aware adjustments, causing modality 3 to be masked less often than the others.

$(x_1^m, x_2^m, \ldots, x_{T_m}^m)$. Each token is a high-dimensional vector obtained through a tokenizer and encodes distinct, non-overlapping information within its modality. These tokens are then fused to produce the final prediction, here using a transformer architecture (Vaswani et al., 2017; Mizrahi et al., 2023). Importantly, we assume that all tokens within a given modality share the same masking probability, denoted $p_m$. Collectively, these form the masking probability vector $\mathbf{p} = (p_1, p_2, \ldots, p_M) \in [0, 1]^M$, which lies in the $M$-dimensional unit hypercube. A masking strategy is then defined as a probability distribution over this hypercube, which may evolve during training.

**Existing masking strategies.** The simplest strategy assigns fixed masking probability to all modalities, e.g., a constant $p_m = 0.75$ (He et al., 2022). To introduce variability in the number of observed tokens, MaskSDM (Zbinden et al., 2026) samples a single masking probability $p \sim \mathcal{U}(0, 1)$ per batch and applies it uniformly: $\mathbf{p} = p \cdot \mathbf{1}_M$. However, when modalities contain multiple tokens, the chance of observing only one modality $m'$ (i.e., all others being fully masked) scales as $\Pi_{m:m \neq m'} p_m^{T_m}$, which decreases exponentially with the number of tokens. Thus, it is highly probable that the model is exposed to at least one token from a dominant modality, making it difficult to learn from other modalities. To encourage more structured input variations, 4M (Mizrahi et al., 2023) samples $\mathbf{p}$ from a symmetric Dirichlet distribution: $\mathbf{p} \sim \text{Dir}(\boldsymbol{\alpha})$ with $\boldsymbol{\alpha} = \mathbf{1}_M$, resulting in uniform sampling over the simplex. However, this constrains the expected proportion of visible tokens to approximately $1/M$ at each iteration, thereby limiting the diversity of the observed input configurations. Modality dropout (Neverova et al., 2015) masks each modality with a fixed probability and, like 4M, treats all modalities equally – thereby ignoring modality competition. OPM (Wei et al., 2024) builds on modality dropout by masking more discriminative modalities more frequently. However, OPM restricts $\mathbf{p} \in \{0, 1\}^M$: each modality is either fully masked or fully visible – preventing partial masking and limiting fine-grained contribution. Moreover, modalities with low performance scores are never masked ($p_m = 0$), while others are masked at near-constant rates (see Appendix A.3). For clarity, the masking strategies discussed above are illustrated in Fig. 3 using a 3-dimensional hypercube (i.e., three modalities).

**Design principles for effective masking.** We identify three key properties that an effective masking strategy over $\mathbf{p} \in [0,1]^M$ should satisfy. (i) **Full support**: the distribution should assign non-zero probability to every $\mathbf{p}$, ensuring that any combination of masked and unmasked tokens can occur. (ii) **Corner prioritization**: points near the corners of the hypercube should be sampled more often, ensuring that the model frequently observes combinations with either almost all tokens or almost none from each modality. In ecological settings, incomplete data often occurs at the modality level rather than at the token level, making full-presence and full-absence cases especially important. In addition, the corners corresponding to $(0, 0, \ldots, 0)$ and $(1, 1, \ldots, 1)$ should be prioritized, as they expose the model to scenarios where either all modalities are available or only a few tokens remain. This helps the model remain reliable both when all modalities are present and when fine-grained contribution analyses require operating with very few tokens. Further motivation for corner prioritization is provided in Appendix A.2.2. (iii) **Imbalance-awareness**: the masking distribution should explicitly address modality imbalance by assigning higher masking probabilities to dominant modalities, which can be identified based on metrics like modality performance or learning speed.

## 3.2 MIAM: Modality Imbalance-Aware Masking

We introduce MIAM, a masking strategy designed to satisfy the three key principles outlined above. First, to ensure full support, we consider the *uniform hypercube* distribution, sampling $\mathbf{p}$ uniformly over the hypercube $[0,1]^M$, i.e., drawing each $p_m \sim \mathcal{U}(0,1)$ independently. Second, to prioritize the corners of the hypercube, we construct a mixture of product beta distributions, each concentrating probability mass around a different corner of the hypercube. This formulation also allows us to increase the likelihood of sampling near the two key corners $(0, 0, \ldots, 0)$ and $(1, 1, \ldots, 1)$, which we call the *beta hypercube* distribution. Finally, MIAM addresses modality imbalance by dynamically adjusting the parameters of the beta hypercube distribution based on modality-specific learning dynamics. We identify dominant modalities by jointly considering their relative performance and learning speed, with the latter estimated as the temporal derivative of performance during training. These modalities are then masked more frequently, encouraging the model to better leverage under-optimized inputs.

**Corner-anchored mixture of product beta distributions.** To construct a flexible, non-uniform probability distribution over the hypercube $[0,1]^M$, we define a mixture of product beta distributions, where each mixture component is designed to concentrate most of its probability mass near one of the $2^M$ corners. Let $\mathrm{Beta}(p_m; \alpha, \beta)$ denote the beta probability density function evaluated at $p_m \in (0,1)$. For a given corner $c = (c_1, \ldots, c_M) \in \{0,1\}^M$ and input $\mathbf{p} = (p_1, \ldots, p_M) \in [0,1]^M$, we define the product beta distribution anchored at corner $c$ as:

$$f_c(\mathbf{p}) = \prod_{m=1}^{M} \begin{cases} \mathrm{Beta}(p_m; 1, \kappa) & \text{if } c_m = 0, \\ \mathrm{Beta}(p_m; \kappa, 1) & \text{if } c_m = 1, \end{cases} \tag{1}$$

where the sharpness parameter $\kappa > 1$ controls the concentration around the corner. We then define the mixture distribution over the set of corners $\mathcal{C} = \{0,1\}^M$:

$$\mathrm{MixProdBeta}(\mathbf{p}) = \sum_{c \in \mathcal{C}} w_c \cdot f_c(\mathbf{p}), \tag{2}$$

where the weights $\{w_c\}_{c \in \mathcal{C}}$ are nonnegative and sum to 1. We allow asymmetric weighting to emphasize specific corners – for example, assigning larger weights to the corners $(0, \ldots, 0)$ and $(1, \ldots, 1)$. Specifically, we set

$$w_c = \begin{cases} \frac{1}{4} & \text{if } c \in \{(0, \ldots, 0), \ (1, \ldots, 1)\}, \\ \frac{1}{2(2^M - 2)} & \text{otherwise.} \end{cases} \tag{3}$$

This weighting allocates half of the mass to the two selected corners and evenly distributes the rest between the remaining $2^M - 2$ corners. In the particular case when each modality has only one token, prioritizing corner $(1, \ldots, 1)$ is meaningless, as it masks all inputs. In this case, we reassign its weight to corner $(0, \ldots, 0)$, yielding $w_{(0,\ldots,0)} = \frac{1}{2}$ while leaving the other weights unchanged.

**Modality imbalance coefficients.** To mitigate modality imbalance during training, we modulate the sharpness parameter $\kappa$ of the corner-anchored beta distribution using two modality-specific factors

$\rho_{s_m}$ and $\rho_{d_m}$. The coefficient $\rho_{s_m}$ is computed from $s_m$, the performance score for modality $m$ when evaluated in isolation on a chosen validation set and metric. In contrast, $\rho_{d_m}$ is calculated from the absolute derivative $d_m$ of $s_m$. Both coefficients are normalized via the geometric mean across modalities:

$$\rho_{s_m} = \frac{s_m}{\left(\prod_{m'=1}^{M} s_{m'}\right)^{1/M}}, \qquad\qquad \rho_{d_m} = \frac{d_m}{\left(\prod_{m'=1}^{M} d_{m'}\right)^{1/M}}. \qquad (4)$$

A high $\rho_{s_m}$ indicates that modality $m$ achieves strong unimodal performance, whereas a high $\rho_{d_m}$ reflects rapid improvement or decline in performance. The ratio $\rho_{s_m}/\rho_{d_m}$ thus guides masking: modalities with high and stable performance (high ratio) are masked more frequently, allowing the model to focus on modalities that are less performant or still learning, while continuing to explore all input combinations during training[1]. We incorporate this adaptive masking into the corner-anchored beta distributions by adjusting $\kappa$ asymmetrically, depending on the corner vector $c$:

$$f_c(\mathbf{p}) = \prod_{m=1}^{M} \begin{cases} \text{Beta}\left(\mathbf{p}_m;\ 1,\ \kappa \cdot \left(\frac{\rho_{s_m}}{\rho_{d_m}}\right)^{-\lambda}\right) & \text{if } c_m = 0. \\ \text{Beta}\left(\mathbf{p}_m;\ \kappa \cdot \left(\frac{\rho_{s_m}}{\rho_{d_m}}\right)^{\lambda},\ 1\right) & \text{if } c_m = 1. \end{cases} \qquad (5)$$

Here, $\lambda > 0$ controls the influence of the imbalance ratio $\frac{\rho_{s_m}}{\rho_{d_m}}$ on the sharpness adjustment. Modalities with higher ratios produce beta distributions more concentrated near 1, increasing the probability of their masking. For intuition, the marginal distribution of MIAM is shown in Appendix A.2.1.

## 4 EXPERIMENTS

### 4.1 EXPERIMENTAL SETUP

We evaluate MIAM's ability to handle incomplete data within and across modalities on two ecological datasets with diverse modality types and configurations: GeoPlant (Picek et al., 2024) for species distribution modeling (SDM) and TaxaBench (Sastry et al., 2025) for multimodal species classification. SDM is a cornerstone ecological task that relates species occurrence records to environmental variables (Elith & Leathwick, 2009), where robustness to missing inputs and fine-grained interpretability are essential (Zbinden et al., 2026). Increasingly, these occurrence records span heterogeneous and often incomplete modalities (e.g., image, audio, geolocation), highlighting the need for models capable of robust multimodal species classification, as evaluated by TaxaBench. A brief overview of each dataset is given here, with full details in Appendix A.1.

**GeoPlant** (Picek et al., 2024) integrates three modalities: tabular environmental predictors, Sentinel-2 satellite imagery, and time series from both climate and Landsat satellite data. The task is to predict the presence or absence of plant species at locations across Europe, formulated as a multi-label classification problem. We use the provided vegetation plot survey labels (presence-absence data) and split the data into training (70%), validation (15%), and test (15%) sets using spatial block cross-validation (Roberts et al., 2017). To mitigate spatial autocorrelation, we employ large block sizes (1°×1°). We retain only species with more than 20 observations and evaluate those with at least one presence record in all three splits, yielding 1783 species. Model performance is assessed using the Area Under the ROC Curve (AUC) averaged across species, the standard metric in SDM. For methods requiring per-modality performance scores (OPM and MIAM), we compute the validation AUC at each epoch, with $\lambda = 3$ and $\kappa = 10$ for MIAM. We tokenize the satellite images at both the patch level (5×5 patches) and the channel level (Red, Green, Blue, NIR) level; the time series at both the year (2000-2018 for climate; 2000-2020 for Landsat) and channel level (4 climatic variables; 6 Landsat bands); and assign one token to each of the 48 tabular variables. The tokenization linearly encodes image patches and time series segments with positional embeddings (Dosovitskiy et al., 2020), while tabular variables are tokenized as in Gorishniy et al. (2022). This fine-grained tokenization enables experiments that isolate the contribution of specific inputs, such as specific tabular variables (e.g., BIO1: the annual mean temperature), groups of tabular variables (e.g., WorldClim), individual years of the climatic time series (e.g., 2018), multi-year segments (e.g., 2000–2018), and specific patches (e.g., the center patch) of the satellite image.

---

[1]A small constant such as $\epsilon = 0.001$ can be added to $s_m, d_m$, and $\rho_{d_m}$ to prevent division by zero.

Table 1: **AUC performance on the GeoPlant test set**. Each column corresponds to a different input subset, showing the performance of each masking strategy on that subset. For each masking strategy, the same trained model is evaluated on all subsets. The best score per subset is written in bold, and the average score across input subsets is reported in the last column.

| Modality | | Partial Unimodal | | | | | Unimodal | | | Bimodal | | | All | Avg. |
|---|---|---|---|---|---|---|---|---|---|---|---|---|---|---|
| Tabular | BIO1 | ✔ | ✔ | | | | ✔ | | | ✔ | ✔ | | ✔ | |
| | WorldClim | | ✔ | | | | ✔ | | | ✔ | ✔ | | ✔ | |
| | Others | | | | | | ✔ | | | ✔ | ✔ | | ✔ | |
| Time series | Clim: 2018 | | | ✔ | ✔ | | | ✔ | | ✔ | | ✔ | ✔ | |
| | Clim: 2000-2018 | | | | ✔ | | | ✔ | | ✔ | | ✔ | ✔ | |
| | Landsat | | | | | | | ✔ | | ✔ | | ✔ | ✔ | |
| Sat. image | Center patch | | | | | ✔ | | | ✔ | | ✔ | ✔ | ✔ | |
| | Others | | | | | | | | ✔ | | ✔ | ✔ | ✔ | |
| Constant | | 68.6 | 82.4 | 84.7 | 86.7 | 55.1 | 83.3 | 90.0 | 63.6 | 90.0 | 83.3 | 89.2 | 87.9 | 80.4 |
| Uniform | | 73.3 | 85.7 | **86.3** | **87.2** | 61.2 | 86.9 | 91.1 | 65.6 | 91.6 | 86.2 | **91.8** | **92.0** | 83.2 |
| Dirichlet | | 65.1 | 82.7 | 77.8 | 86.8 | 54.9 | 87.5 | 91.1 | 58.2 | 91.8 | 88.6 | 91.7 | 91.4 | 80.6 |
| Modality dropout | | 48.7 | 80.8 | 77.4 | 86.4 | 66.2 | 88.6 | **91.4** | 73.2 | **92.0** | 89.2 | 91.7 | **92.0** | 81.5 |
| OPM | | 68.0 | 81.9 | 80.7 | 85.3 | 68.1 | 88.4 | 90.2 | **81.1** | 90.7 | **89.5** | 91.1 | 91.2 | 83.8 |
| MIAM (ours) | | **78.4** | **86.7** | 86.0 | 87.0 | **70.8** | **89.0** | 91.4 | 80.1 | 91.7 | **89.5** | 91.5 | 91.7 | **86.1** |
| Oracle (one model per column) | | 78.0 | 87.1 | 87.7 | 87.6 | 77.1 | 89.3 | 92.2 | 81.4 | 92.3 | 89.7 | 91.7 | 92.0 | 87.2 |

**TaxaBench** (Sastry et al., 2025) consists of species observations sourced from iNaturalist (Van Horn et al., 2018), each associated with five modalities: a ground-level image, a satellite image, an audio recording, environmental tabular predictors, and the geolocation. The task is to classify the species represented in each sample. Since the dataset was originally designed for zero-shot classification, no predefined splits are available. We therefore retain only species with at least 10 observations (199 species) and stratify the remaining 4876 samples into train (80%), validation (10%), and test (10%) sets, ensuring the same proportions of species in all splits. The validation loss at each epoch is used to compute per-modality scores, and we set $\lambda = 1$ and $\kappa = 10$ for MIAM. We reuse the pre-trained encoders from Sastry et al. (2025), where each encoder outputs a single token per modality. The resulting five tokens are then masked according to the masking strategy and passed to the transformer for the final prediction. Since we consider only one token per modality, the constant and modality dropout masking strategies are equivalent.

In Section 4.2.1, we compare MIAM against existing masking strategies and an *oracle* baseline: a non-masking model trained directly on the exact subset of tokens under evaluation. Since the number of possible subsets grows exponentially, $O(2^{\sum_{m=1}^{M} T_m})$, oracle models are impractical at scale. We therefore include them only as an approximate upper bound on performance for specific input subsets and under the given model and training setup. When restricted to a single modality, the oracle reduces to a unimodal baseline. All models and baselines are trained under the same protocol for a given dataset (detailed in Appendix A.1 and A.3), differing only in their masking strategies. For each masking strategy, the same trained model is evaluated across all input subsets, meaning that no retraining is performed when switching subsets. An ablation and sensitivity analysis of MIAM are presented in Section 4.2.2 and Appendix A.4.1.

## 4.2 RESULTS

### 4.2.1 COMPARISON AMONG MASKING STRATEGIES

Table 1 shows the AUC performance of the different masking strategies on the GeoPlant dataset. Overall, MIAM outperforms all baselines, with an average gain of 2.3% over the second-best method. MIAM performs strongly across all unimodal setups, substantially narrowing the gap caused by modality imbalance, which is most evident for satellite imagery. OPM performs slightly better on satellite imagery and on some of the subsets it favors during training (e.g., satellite & tabular or satellite & time series). However, it fails considerably on subsets it never encounters, particularly partial unimodal setups, where MIAM remains robust and uniform masking is the main

Table 2: **Top-1 accuracy on the test set of TaxaBench**. Each column corresponds to a different input subset, showing the performance of each masking strategy on that subset. For each masking strategy, the same trained model is evaluated on all subsets. The best score per subset is highlighted in bold, and the average performance for each masking strategy across input subsets is reported in the last column.

| Modality | Input Type | | | | | | | | | | | | |
|---|---|---|---|---|---|---|---|---|---|---|---|---|---|
| | Unimodal | | | | | Bimodal | | Trimodal | | Quadri. | | All | Avg. |
| Ground-level image | ✔ | | | | | ✔ | ✔ | ✔ | | ✔ | | ✔ | |
| Audio | | ✔ | | | | ✔ | | ✔ | | | ✔ | ✔ | |
| Geographic location | | | ✔ | | | | ✔ | ✔ | ✔ | ✔ | ✔ | ✔ | |
| Environmental features | | | | ✔ | | | | | ✔ | ✔ | ✔ | ✔ | |
| Satellite image | | | | | ✔ | | | | ✔ | ✔ | ✔ | ✔ | |
| Uniform | **42.4** | 41.2 | **8.40** | **7.99** | 6.76 | 59.2 | 48.8 | 64.3 | 9.02 | 51.2 | 46.9 | 65.8 | 37.7 |
| Dirichlet | 42.2 | 40.8 | 5.33 | 5.12 | 7.58 | 59.2 | 48.4 | 65.0 | 9.63 | 51.4 | 45.9 | 67.8 | 37.4 |
| Modality dropout | 41.4 | 39.8 | 5.53 | 4.51 | 8.2 | 57.2 | 44.3 | 59.2 | 9.63 | 51.0 | 45.1 | 65.0 | 35.9 |
| OPM | 33.2 | 35.0 | 5.74 | 5.12 | 7.79 | 46.3 | 34.4 | 50.0 | **10.9** | 43.6 | 42.6 | 59.4 | 31.2 |
| MIAM (ours) | 42.2 | **41.8** | 6.56 | 7.38 | **9.84** | **60.9** | **50.2** | **65.4** | 10.2 | **52.0** | **49.0** | **69.1** | **38.7** |
| Oracle (one model per column) | 45.3 | 44.9 | 7.58 | 9.43 | 12.9 | 63.3 | 50.0 | 66.6 | 13.1 | 51.8 | 46.5 | 69.1 | 40.0 |

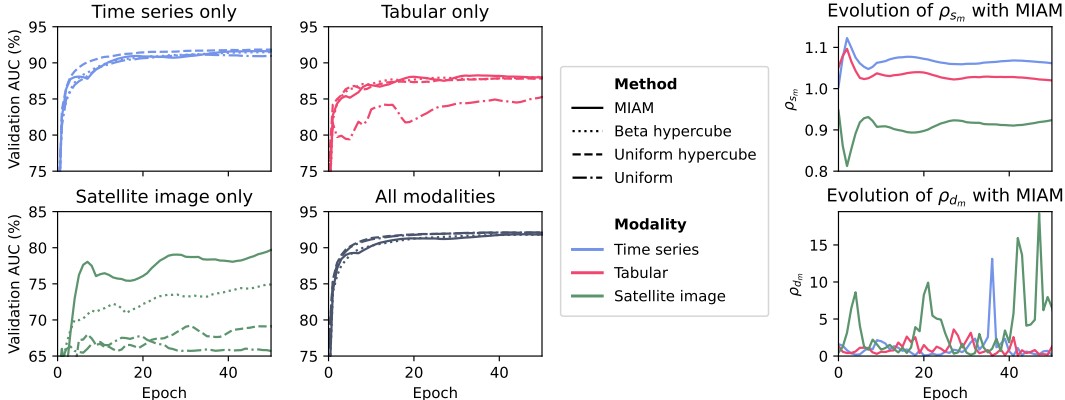

Figure 4: **Ablation study and effect of dynamic masking.** Validation AUC on GeoPlant for each modality and intermediate masking strategies leading to MIAM (left), and evolution of MIAM's modality imbalance coefficients $\rho_{s_m}$ and $\rho_{d_m}$ during training (right).

competitor. Yet, uniform masking does not account for modality dominance, resulting in poor performance on satellite imagery. When using all modalities, MIAM performs slightly below modality dropout and uniform masking. However, this gap is eliminated by reducing the strength of the imbalance-aware coefficients via $\lambda$, which controls the tradeoff between dominant and dominated modalities (see Appendix A.4.1 for our hyperparameter analysis). Finally, the gap between MIAM and the oracle baseline is small, indicating that MIAM approaches the performance of specialized models trained directly on each input subset, while remaining broadly flexible.

Table 2 reports the top-1 accuracy of different masking strategies on the TaxaBench dataset. On average, MIAM achieves the strongest performance, particularly in multimodal settings. The only exception is the trimodal case excluding the dominant ground-level image and audio, where OPM performs better since this subset appears most frequently during its training. In unimodal cases, both MIAM and uniform masking remain strong baselines. Additional metrics (top-5 accuracy and F1-score) are reported in Appendix A.4.2 and show the same trends as the top-1 accuracy.

### 4.2.2 ABLATION STUDY

In the left side of Fig. 4, we show validation performance on GeoPlant during training for uniform masking and the successive variants of MIAM: starting with the uniform hypercube (full-support

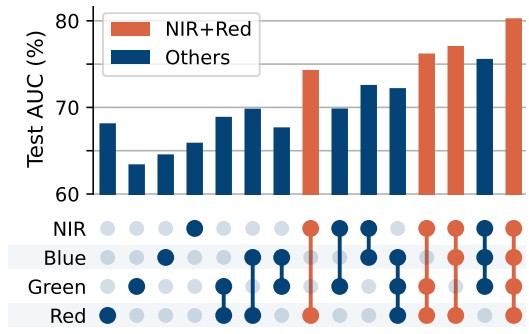
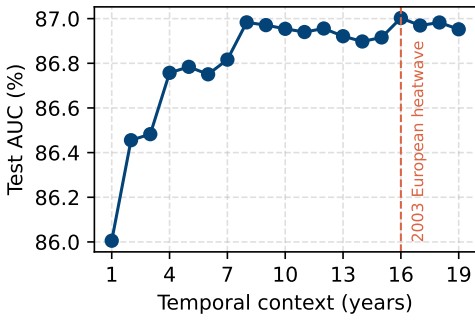

(a) Performance across different spectral band combinations of satellite imagery

(b) Performance with increasing historical climatic context

Figure 5: **Contribution analysis for ecological insights.** MIAM enables fine-grained contribution within modalities. **(a)** The Red and NIR bands are particularly important as they are used to compute vegetation indices such as NDVI. **(b)** Extending the temporal context (longer time series) captures signals from past extreme events, such as heatwaves.

principle), extending to the beta hypercube (corner-prioritization principle), and finally to MIAM (imbalance-aware principle). While performance is similar for the dominant time series modality and for all modalities combined, each added principle consistently improves results on the other modalities, most notably for satellite imagery. MIAM also displays cyclic patterns, driven by fluctuations in the relative learning speed $\rho_{d_m}$, which periodically shift the training focus across modalities. This link is evident in the correspondence between the $\rho_d$ curve (right side of Fig. 4) and the validation performance curve of MIAM. We hypothesize that such cyclic effects are beneficial for learning, similar to periodic learning-rate schedules. By contrast, the relative performance scores $\rho_{s_m}$ remain fairly stable throughout training: while they provide a useful prior for identifying dominant modalities, relying solely on them, as in OPM, leads to suboptimal performance because the masking distribution evolves little over time (see Appendix A.4.1 for numerical results).

Table 3 illustrates the positive effect of using non-uniform corner weights $w_c$, which prioritize the corners $(0, 0, \ldots, 0)$ and $(1, 1, \ldots, 1)$. The table reports the average performance gain across the input subsets of Table 1 and Table 2 on their validation sets. While the improvement is modest on GeoPlant, the gain on TaxaBench is more pronounced – potentially because it includes more modalities, making these key corners occur even less frequently under uniform corner weights. Additional ablation studies and a sensitivity analysis of the hyperparameters are provided in Appendix A.4.1.

Table 3: Average performance impact of the prioritization of key corners by using a non-uniform $w_c$, evaluated on the validation sets of GeoPlant (AUC) and TaxaBench (Top-1 accuracy).

|  | GeoPlant | TaxaBench |
|---|---|---|
| Uniform $w_c$ | 85.2 | 36.0 |
| Non-uniform $w_c$ | **85.4** | **37.1** |

## 5 DISCUSSION

The strong performance of MIAM on incomplete multimodal data, both across and within modalities, enables more accurate estimation of input contributions. By comparing performance across modalities or subsets of tokens, we can identify which inputs drive performance and may play critical ecological roles. For example, in the GeoPlant dataset, we find that model performance consistently improves when the Red and NIR spectral bands of satellite imagery are used together (Fig. 5a). This aligns with well-established knowledge in remote sensing and vegetation monitoring: Red and NIR bands are used to compute the Normalized Difference Vegetation Index (NDVI), which captures biomass and phenological patterns of vegetation (Pettorelli et al., 2005; He et al., 2015). In addition, our results indicate that model performance generally improves with longer time series (Fig. 5b). We observe a significant increase in performance when the time series includes the 2003 European heatwave, highlighting the importance of including temporal scales that capture key

information such as extreme events (Lynch et al., 2014; Fonteyn et al., 2025). However, analyses such as in Fig. 5a rely on manual inspection of selected subsets, which becomes impractical as the number of tokens grows. To obtain systematic and interpretable measures, as future work, we plan to compute token-level Shapley values, yielding a single contribution score per token, thus extending prior approaches to multimodal settings (Covert et al., 2023; Zbinden et al., 2026).

Additionally, the formulation of masking for multimodal learning developed in this work can serve as a foundation for designing new strategies tailored to specific multimodal learning challenges. For instance, if particular token subsets are of higher importance than the others, the corner weights $w_c$ can be adjusted accordingly. MIAM can also be beneficial for large multimodal models such as 4M (Mizrahi et al., 2023), which currently rely on the Dirichlet distribution and may suffer from modality imbalance during training. In particular, its advantages are most evident in datasets with many modalities and more than one token per modality. Our experiments on the bimodal SatBird dataset (Teng et al., 2023) show that, in such a simple setting, all masking strategies perform similarly (see Appendix A.4.3). This highlights the need for richer multimodal benchmarks, both in ecology and in machine learning more broadly, that incorporate more than two modalities.

Finally, while we focus on supervised setups in this work, masking strategies are often used for self-supervised pre-training. However, to the best of our knowledge, modality imbalance has not been explored in the context of SSL. To assess the potential of MIAM in an SSL setting, we conduct a small-scale, proof-of-concept experiment. Although the Beta hypercube variant (MIAM with $\lambda = 0$) can be implemented directly, MIAM requires per-modality performance scores, which are difficult to obtain in SSL because labels are absent. For this experiment, we estimate modality performance using the reconstruction losses computed on the training set.

While alternative approaches could be explored, this choice is simple and broadly applicable. The experimental design follows the MultiMAE framework (Bachmann et al., 2022), adapted to the modalities in GeoPlant. The model is pre-trained on the training set using a given masking strategy, linear weights for linear probing (LP) are learned on the validation set, and evaluation across input subsets is performed on the test set. Full experimental details and per-subset results are provided in Appendix A.4.4, with a summary shown in Table 4. Overall, MIAM outperforms the other masking strategies under LP, demonstrating its promise for self-supervised settings, with the uniform strategy (Zbinden et al., 2026) performing comparably. Although this is a small-scale experiment, it highlights the importance of exploring masking strategies in self-supervised multimodal learning and suggests that the commonly used Dirichlet distribution may not be optimal, given MIAM's consistent improvements.

Table 4: Average AUC performance of masking strategies on GeoPlant with SSL pre-training evaluated via linear probing. Per-subset results are provided in Table 10.

| Method | Avg. |
|---|---|
| Constant | 78.3 |
| Uniform | 79.3 |
| Dirichlet | 77.0 |
| Modality dropout | 77.5 |
| OPM | 75.3 |
| MIAM (ours) | **79.5** |

## 6 CONCLUSION

Leveraging multimodal learning with masking strategies is essential in ecological applications, where data is heterogeneous, incomplete, and requires interpretable input contributions. By formalizing masking strategies for multimodal settings, we introduce MIAM, a dynamic masking strategy that effectively covers the space of input subsets while addressing modality imbalance, a central challenge in multimodal learning. Our results demonstrate that MIAM consistently outperforms existing masking approaches across multiple ecological datasets, enabling models to handle arbitrary input subsets and yielding valuable ecological insights through fine-grained contribution analysis.

### REPRODUCIBILITY STATEMENT

We include all necessary details to ensure reproducibility. Model architectures and experimental setups are presented in the main text and the appendix. The code and trained model weights are publicly available at `https://github.com/zbirobin/MIAM` and `https://huggingface.co/zbirobin/MIAM`.

ACKNOWLEDGMENTS

This work was supported by the Swiss National Science Foundation under grant 200021_204057 "Learning unbiased habitat suitability at scale with AI (deepHSM)," and by the Horizon Europe grant 101213369 (DVPS). We thank Julie Charlet for her early contributions to the time series component and Emanuele Dalsasso for his precious feedback.

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

## A    APPENDIX

The appendix contains the following information:

- Datasets and training details (A.1)
- MIAM marginal distributions and the rationale behind corner prioritization (A.2)
- Baselines details (A.3)
- Additional experiments (A.4): ablation study and hyperparameter analysis, extra metrics on TaxaBench, results on SatBird, and an evaluation of MIAM in a self-supervised learning pre-training context
- LLM usage (A.5)

### A.1    DATASETS AND TRAINING DETAILS

We provide additional information on the datasets and their associated training procedures. All baselines share the same training protocol and model architecture; the only difference lies in the masking strategy. The following details are common to both GeoPlant and TaxaBench. Models are trained with AdamW (Loshchilov & Hutter, 2017) using a weight decay of 0.01, a schedule-free approach (Defazio et al., 2024), learning rate of 0.001, batch size of 128, and dropout rate of 0.1 (Srivastava et al., 2014). Training runs for 100 epochs with early stopping based on the average validation AUC across the unimodal setups and the multimodal setup with all modalities included.

The models consist of tokenizers that produce tokens for each modality, followed by a transformer (Vaswani et al., 2017) that fuses them. Tokens are masked according to the selected strategy, with masked tokens replaced by a learned mask embedding, and the resulting sequence is processed by a 3-block transformer with 8 heads, similar to (Gorishniy et al., 2021). The transformer outputs the same number of tokens, which are averaged to obtain a final representation and then passed through a linear layer to produce the logits. The next two sections provide details specific to each dataset.

#### A.1.1    GEOPLANT

The geographic distribution of the data splits is shown in Fig. 6, and the code to reproduce the exact splits is provided in Fig. 7, using the `verde`[2] library for block cross-validation (Roberts et al., 2017).

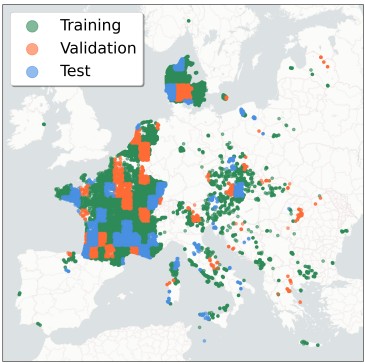

Figure 6: **GeoPlant geographic distribution of the data splits**. Blocks are of size $1° \times 1°$.

We tokenize each input part (patch, time segment, or tabular variable) independently. This yields 76 tokens for the climate time series (4 channels $\times$ 19 years), 126 tokens for the Landsat time series (6 channels $\times$ 21 years), 100 tokens for Landsat patches (4 channels $\times$ 25 patches), and 48 tokens for the 48 tabular variables. Patches and time segments are encoded following Dosovitskiy et al. (2020), using a linear projection to match the token dimension along with sinusoidal positional embeddings.

---

[2]https://www.fatiando.org/verde/latest/

```python
import pandas as pd
import verde as vd

df = pd.read_csv("/path/to/geoplant/PA_metadata_train.csv")
coordinates = df[["lat", "lon"]].to_numpy()
spacing: float = 1.
test_size: float = 0.15
val_size: float = 0.15

train_block, test_block = vd.train_test_split(
    coordinates.transpose(),
    df.index.to_numpy(),
    spacing=spacing,
    test_size=test_size,
    random_state=42,
)
train_indices, test_indices = train_block[1][0], test_block[1][0]

train_block, val_block = vd.train_test_split(
    coordinates[train_indices].transpose(),
    train_indices,
    spacing=spacing,
    test_size=val_size / (1 - test_size),
    random_state=42,
)
train_indices, val_indices = train_block[1][0], val_block[1][0]
```

Figure 7: Python code for splitting the GeoPlant data using the `verde` library for block cross-validation (Roberts et al., 2017).

```python
import pandas as pd
from sklearn.model_selection import train_test_split
from sklearn.preprocessing import LabelEncoder

df = pd.read_csv("/path/to/taxabench/test_df.csv")
df = df[
    df["scientific_name"].isin(
        pd.DataFrame(df.value_counts(subset="scientific_name"))
        .query("count >= 10")
        .index
    )
]
y = LabelEncoder().fit_transform(df["scientific_name"])
X = df.drop(columns="scientific_name")

X_train, X_remainder, y_train, y_remainder = train_test_split(
    X, y, test_size=0.2, stratify=y, random_state=42,
)
X_test, X_val, y_test, y_val = train_test_split(
    X_remainder, y_remainder, test_size=0.5,
    stratify=y_remainder, random_state=42,
)
```

Figure 8: Python code for stratified splitting of species observations of TaxaBench.

Tabular variables are tokenized as in Gorishniy et al. (2021) by projecting each scalar into a higher-dimensional space with periodic activation functions. All tokens have size 192. A final sigmoid is applied to the logits to predict the presence of 1783 species. The model is trained as a multi-label classifier using the weighted binary cross-entropy loss from Zbinden et al. (2024).

### A.1.2 TAXABENCH

The code to generate the stratified splits is provided in Fig. 8. We reuse the pre-trained encoders released by the dataset authors (Sastry et al., 2025). Each encoder outputs a single token of size 512 per modality, resulting in a total of five tokens. A softmax function is applied to the final logits, and the model is trained with the cross-entropy loss.

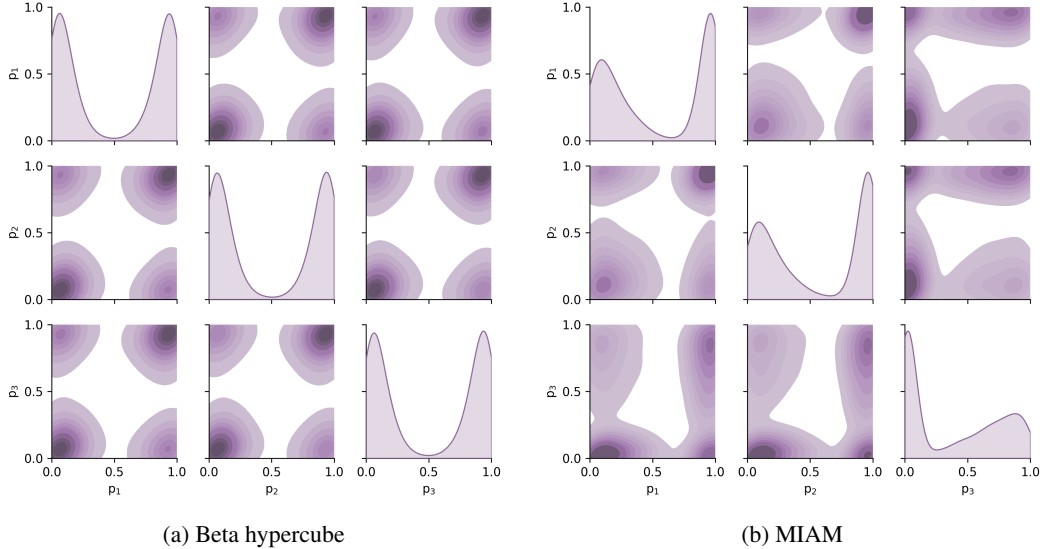

(a) Beta hypercube                          (b) MIAM

Figure 9: **Marginal distributions of the beta hypercube and MIAM.** Same setup as in Fig. 3, with three modalities where modality 3 is dominated, but here showing the marginal distributions of the beta hypercube and MIAM. Each panel shows the kernel density estimate of 5000 sampled points. **(a)** Beta hypercube: a mixture of product beta distribution to prioritize corners **(b)** MIAM (dynamic masking): a beta hypercube with imbalance-aware adjustments, causing modality 3 to be masked less frequently than the others.

## A.2    ADDITIONAL MIAM INTUITIONS

### A.2.1    MIAM MARGINAL DISTRIBUTIONS

We provide an additional visualization in Fig. 9 showing the marginal distributions of the beta hypercube and MIAM, which offers further intuition into how MIAM operates.

### A.2.2    CORNER PRIORITIZATION

The motivation for prioritizing the corners and for overweighting certain corners more than others stems from both ecological considerations and optimization behavior:

- **Prioritizing corners**: In ecological applications, incomplete data often occurs at the level of entire modalities (e.g., a satellite image at a given timestamp). Prioritizing corners reflects this situation by emphasizing full-presence or full-absence cases. Moreover, tokens from the same modality tend to be highly correlated (e.g., spectral bands, temporal segments), meaning that observing only a few tokens may already capture most of the modality's information. This can exacerbate modality imbalance if multiple tokens of the dominant modalities are always present, further motivating the need to emphasize complete absence cases.

- **All-tokens corner**: It is also common in ecological datasets to have (nearly) complete observations across all modalities. Ensuring strong performance in these frequent, fully observed cases is crucial for overall accuracy and generalization. Prioritizing this corner also encourages the model to learn joint cross-modal interactions efficiently when all inputs are available, leveraging complementary information across modalities.

- **Few-tokens corner**: Given the high intra-modality correlation, it is important that the model remains capable of extracting useful information even when only a few tokens are available. This is particularly relevant for interpretability: if we wish to assess the contribution of a single token (e.g., a specific tabular variable), the model must have encountered situations in which the corresponding token appeared nearly alone during training.

Table 5: **Ablation of MIAM components.** Validation AUC on GeoPlant when different elements of MIAM are removed. Each column corresponds to a different input subset, showing the performance of each masking strategy on that subset. For each masking strategy, the same trained model is evaluated on all subsets. The best score per subset is shown in bold and the average performance across all input subsets is reported in the final column.

| Modality | | Partial Unimodal | | | | | Unimodal | | | Bimodal | | | All | Avg. |
|---|---|---|---|---|---|---|---|---|---|---|---|---|---|---|
| **Tabular** | BIO1 | ✔ | ✔ | | | | ✔ | | | ✔ | ✔ | | ✔ | |
| | WorldClim | | ✔ | | | | ✔ | | | ✔ | ✔ | | ✔ | |
| | Others | | | | | | ✔ | | | ✔ | ✔ | | ✔ | |
| **Time series** | Clim: 2018 | | | ✔ | ✔ | | | ✔ | | ✔ | | ✔ | ✔ | |
| | Clim: 2000-2018 | | | | ✔ | | | ✔ | | ✔ | | ✔ | ✔ | |
| | Landsat | | | | | | | ✔ | | ✔ | | ✔ | ✔ | |
| **Sat. image** | Center patch | | | | | ✔ | | | ✔ | | ✔ | ✔ | ✔ | |
| | Others | | | | | | | | ✔ | | ✔ | ✔ | ✔ | |
| MIAM | | **73.1** | 85.3 | **85.5** | 86.0 | **70.9** | **88.4** | 91.6 | **80.1** | **88.9** | 91.7 | 91.9 | 91.7 | **85.4** |
| MIAM w/o $\rho_{s_m}$ | | 67.7 | **85.4** | 85.0 | **86.2** | 70.1 | 88.0 | 91.5 | **80.1** | 88.8 | 91.4 | 91.7 | 91.5 | 84.8 |
| MIAM w/o $\rho_{d_m}$ | | 71.3 | **85.4** | 85.2 | 86.1 | 70.1 | 87.8 | **91.8** | 76.4 | 88.7 | **91.9** | **92.0** | **92.0** | 84.9 |
| MIAM w/ uniform $w_c$ | | 73.0 | 85.2 | 85.3 | 85.9 | 70.6 | 88.3 | 91.4 | 79.5 | **88.9** | 91.3 | 91.6 | 91.4 | 85.2 |

While these motivations are partly hypothesis-driven, we observe empirical support for them. Fig. 4 highlights the performance impact on dominated modalities (high difference between Beta and Uniform hypercubes), and the ablation study in Table 3 shows consistent performance gains – especially for dominated modalities like satellite imagery (see Table 5) – when using non-uniform corners weights $w_c$. Importantly, as mentioned in the discussion, MIAM remains flexible: for different applications, $w_c$ can be adapted to emphasize alternative corners that better reflect application-specific data patterns.

## A.3 BASELINES DETAILS

We provide additional details on the baselines and explain why OPM constitutes a suboptimal masking strategy.

- **Constant**: the probability of masking each token is fixed at $0.75$, following He et al. (2022).
- **Uniform**: at every iteration, a shared masking probability $p \sim \mathcal{U}(0, 1)$ is sampled and applied across modalities.
- **Dirichlet**: following Mizrahi et al. (2023), we use a symmetric Dirichlet with $\boldsymbol{\alpha} = \mathbf{1}_M$.
- **Modality dropout**: following Neverova et al. (2015), all tokens from a given modality are masked with probability $0.1$.
- **OPM**: we follow the definition from Wei et al. (2024) and adopt their hyperparameters, $q_{base} = 0.5$ and $\lambda = 0.4$. In practice, this formulation produces largely fixed masking behavior during training, as per-modality performance scores remain nearly constant. Relying solely on these scores can also be misleading, since low-performing modalities may simply be uninformative rather than worth emphasizing. Another limitation is that dominated modalities are never masked, which prevents the model from being exposed to the full range of modality combinations. Finally, OPM, as originally defined, only drops entire modalities, without enabling fine-grained masking within them.

## A.4 ADDITIONAL EXPERIMENTS

### A.4.1 ABLATION AND SENSITIVITY ANALYSIS

Table 5 highlights the importance of incorporating both modality imbalance coefficients, $\rho_{s_m}$ and $\rho_{d_m}$: removing either reduces performance by at least 0.5% on average. In particular, dropping $\rho_{d_m}$ substantially degrades the unimodal setup on satellite imagery, underscoring its role in identifying

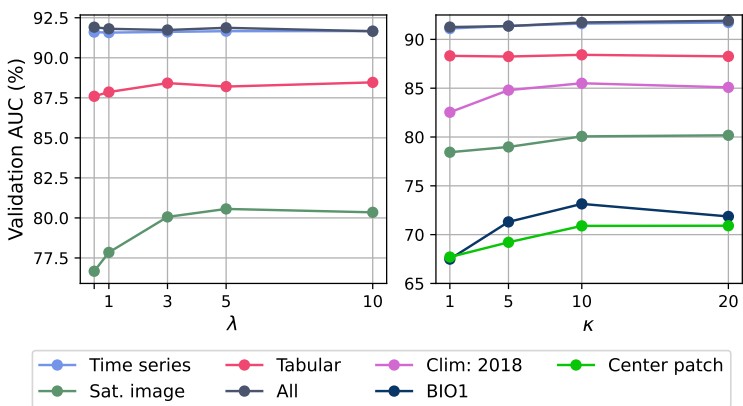

Figure 10: **Sensitivity of MIAM to $\lambda$ and $\kappa$ on the GeoPlant dataset.** The reported metric is the validation AUC as $\lambda$ and $\kappa$ are varied.

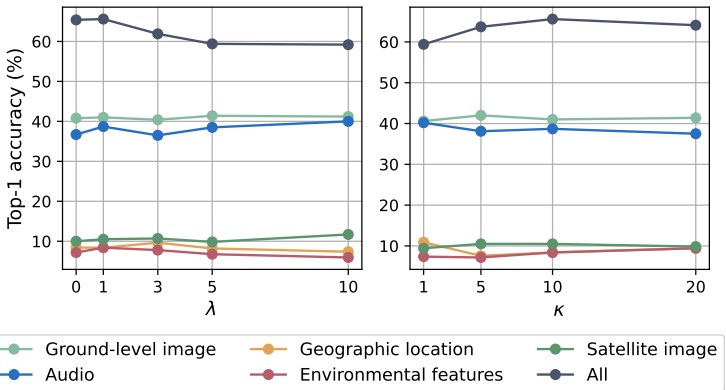

Figure 11: **Sensitivity of MIAM to $\lambda$ and $\kappa$ on the TaxaBench dataset.** The reported metric is the validation Top-1 accuracy as $\lambda$ and $\kappa$ are varied.

dominated modalities. The same figure also shows a positive effect from using non-uniform corner weights $w_c$, which prioritize the corners $(0, 0, \ldots, 0)$ and $(1, 1, \ldots, 1)$ and yield a modest but consistent improvement across all subsets.

MIAM depends on the hyperparameters $\lambda$ and $\kappa$. In Fig. 10, we observe that $\lambda$ controls the tradeoff between dominant and dominated modalities on GeoPlant, with larger values improving the performance of dominated modalities, i.e., satellite imagery. The same figure demonstrates that $\kappa$ also plays an important role, particularly for setups with fewer tokens, where we see more variation in performance depending on its value. However, a similar experiment on the TaxaBench dataset (Fig. 11) indicates that these hyperparameters are less critical in some settings (notably TaxaBench, which contains only one token per modality), as performance remains stable across different values. We recommend setting $\kappa = 10$ and choosing $\lambda$ between 1 and 5, depending on the degree of modality imbalance. Note that because $\lambda$ and $\kappa$ parameterize the masking distribution, their effects can be inspected directly by varying them and visualizing the resulting distributions (e.g., Fig. 3 and Fig. 9), without requiring model retraining. Naturally, they can also be fine-tuned like any other hyperparameter using a validation set.

In Table 6, we evaluate the masking strategies using higher-capacity models to assess whether our findings hold at larger scales. We consider a relatively large transformer variant (6 layers, 256-dimensional tokens) and a relatively huge variant (12 layers, 512-dimensional tokens), and evaluate the masking strategies on them. Across both larger models, MIAM remains the strongest masking strategy on average, outperforming the other baselines by a clear margin. However, we also observe a decline in overall performance as model size increases, which may indicate overfitting and suggest that additional data would be needed to fully leverage these larger models.

Table 6: **AUC performance on the GeoPlant test set with increased model size**. Model size is varied by adjusting the number of layers and the token dimension; the base configuration is the one used throughout the main paper (see Table 1). Each column corresponds to a different input subset, showing the performance of each masking strategy on that subset. For each masking strategy, the same trained model is evaluated on all subsets. The best score per subset is shown in bold, the best score per subset within each model size setting is underlined, and the average performance across all input subsets is reported in the final column.

| Modality | | Partial Unimodal | | | | | Unimodal | | | Bimodal | | | All | Avg. |
|---|---|---|---|---|---|---|---|---|---|---|---|---|---|---|
| **Tabular** | BIO1 | ✔ | ✔ | | | | ✔ | | | ✔ | ✔ | | ✔ | |
| | WorldClim | | ✔ | | | | ✔ | | | ✔ | ✔ | | ✔ | |
| | Others | | | | | | ✔ | | | ✔ | ✔ | | ✔ | |
| **Time series** | Clim: 2018 | | | ✔ | ✔ | | | ✔ | | ✔ | | ✔ | ✔ | **Avg.** |
| | Clim: 2000-2018 | | | | ✔ | | | ✔ | | ✔ | | ✔ | ✔ | |
| | Landsat | | | | | | | ✔ | | ✔ | | ✔ | ✔ | |
| **Sat. image** | Center patch | | | | | ✔ | | | ✔ | | ✔ | ✔ | ✔ | |
| | Others | | | | | | | | ✔ | | ✔ | ✔ | ✔ | |
| **Base**: 3 layers, 192 dimensions | | | | | | | | | | | | | | |
| Constant | | 68.6 | 82.4 | 84.7 | 86.7 | 55.1 | 83.3 | 90.0 | 63.6 | 90.0 | 83.3 | 89.2 | 87.9 | 80.4 |
| Uniform | | 73.3 | 85.7 | 86.3 | 87.2 | 61.2 | 86.9 | 91.1 | 65.6 | 91.6 | 86.2 | 91.8 | **92.0** | 83.2 |
| Dirichlet | | 65.1 | 82.7 | 77.8 | 86.8 | 54.9 | 87.5 | 91.1 | 58.2 | 91.8 | 88.6 | 91.7 | 91.4 | 80.6 |
| Modality dropout | | 48.7 | 80.8 | 77.4 | 86.4 | 66.2 | 88.6 | 91.4 | 73.2 | **92.0** | 89.2 | 91.7 | **92.0** | 81.5 |
| OPM | | 68.0 | 81.9 | 80.7 | 85.3 | 68.1 | 88.4 | 90.2 | **81.1** | 90.7 | 89.5 | 91.1 | 91.2 | 83.8 |
| MIAM (ours) | | **78.4** | **86.7** | 86.0 | 87.0 | **70.8** | **89.0** | 91.4 | 80.1 | 91.7 | 89.5 | 91.5 | 91.7 | **86.1** |
| **Large**: 6 layers, 256 dimensions | | | | | | | | | | | | | | |
| Constant | | 58.0 | 82.2 | 83.9 | 86.8 | 55.1 | 81.4 | 89.6 | 55.4 | 89.9 | 81.2 | 90.5 | 90.6 | 78.7 |
| Uniform | | 68.7 | 84.5 | **86.4** | 86.9 | 65.3 | 86.1 | 90.4 | 66.6 | 90.7 | 86.1 | 91.4 | 91.6 | 82.9 |
| Dirichlet | | 69.1 | 84.3 | 79.2 | 85.6 | 53.5 | 87.6 | 91.2 | 59.3 | 92.0 | 88.5 | 92.0 | 92.0 | 81.2 |
| Modality dropout | | 66.0 | 81.9 | 74.5 | 86.1 | 65.0 | 88.4 | 91.5 | 73.7 | 91.9 | 88.9 | 91.8 | 91.9 | 82.6 |
| OPM | | 62.0 | 82.4 | 68.1 | 86.1 | 69.5 | 86.8 | 88.6 | 80.0 | 88.8 | 89.0 | 90.1 | 90.2 | 81.8 |
| MIAM (ours) | | 74.5 | 86.0 | 85.0 | **87.3** | 70.3 | 88.5 | 90.7 | 78.7 | 90.9 | 89.2 | 90.9 | 90.9 | 85.2 |
| **Huge**: 12 layers, 512 dimensions | | | | | | | | | | | | | | |
| Constant | | 58.6 | 72.0 | 84.5 | 85.5 | 57.8 | 75.2 | 88.1 | 61.1 | 88.3 | 78.3 | 88.8 | 88.4 | 77.2 |
| Uniform | | 69.6 | 77.4 | 84.2 | 85.6 | 62.4 | 81.8 | 86.5 | 64.3 | 87.3 | 83.7 | 88.9 | 89.2 | 80.1 |
| Dirichlet | | 57.4 | 79.1 | 81.5 | 86.3 | 60.9 | 83.8 | 89.9 | 62.1 | 90.6 | 85.5 | 91.0 | 91.2 | 79.9 |
| Modality dropout | | 54.0 | 77.8 | 72.8 | 86.4 | 65.0 | 87.0 | 88.7 | 70.5 | 89.8 | 88.0 | 89.7 | 90.3 | 80.0 |
| OPM | | 62.1 | 81.0 | 73.7 | 81.5 | 66.0 | 84.4 | 82.9 | 75.4 | 84.9 | 87.1 | 86.9 | 87.5 | 79.4 |
| MIAM (ours) | | 63.8 | 82.7 | 83.0 | 85.9 | 67.6 | 84.9 | 86.8 | 75.4 | 86.9 | 87.4 | 88.6 | 88.8 | 81.8 |

### A.4.2 ADDITIONAL METRICS FOR TAXABENCH

We provide additional metrics for TaxaBench – top-5 accuracy (Table 7) and F1-Score (Table 8) – complementing the top-1 accuracy shown in the main text. Interestingly, across all three metrics, masking strategies outperform the oracle on certain modality subsets. We hypothesize that this occurs because the tokens, obtained from the pre-trained encoders of Sastry et al. (2025), are already well aligned, allowing masking strategies to exploit cross-modal features by exposing the model to more tokens. By contrast, the oracle is limited to tokens from the evaluated subset and cannot benefit from this cross-modal information. In addition, the relatively small dataset makes results more variable, and masking may provide a regularizing effect that improves generalization.

### A.4.3 SATBIRD

We explore the SatBird dataset (Teng et al., 2023), a benchmark for species distribution models with only two modalities: tabular environmental predictors and Sentinel-2 satellite imagery. With just two modalities, the performance differences between baselines are minimal, making the dataset less useful for evaluating masking strategies. Nevertheless, we include it here for completeness and transparency.

Table 7: **Top-5 accuracy on the test set of TaxaBench**. Each column corresponds to a different input subset, showing the performance of each masking strategy on that subset. For each masking strategy, the same trained model is evaluated on all subsets. The best score per subset is written in bold, and the average score across input subsets is reported in the last column.

| Modality | Input Type | | | | | | | | | | | | Avg. |
| --- | --- | --- | --- | --- | --- | --- | --- | --- | --- | --- | --- | --- | --- |
| | Unimodal | | | | | Bimodal | | Trimodal | | Quadri. | | All | |
| Ground-level image | ✔ | | | | | ✔ | ✔ | ✔ | | ✔ | | ✔ | |
| Audio | | ✔ | | | | ✔ | | ✔ | | | ✔ | ✔ | |
| Geographic location | | | ✔ | | | | ✔ | ✔ | ✔ | ✔ | ✔ | ✔ | |
| Environmental features | | | | ✔ | | | | | ✔ | ✔ | ✔ | ✔ | |
| Satellite image | | | | | ✔ | | | | | ✔ | ✔ | ✔ | |
| Uniform | 65.6 | **67.4** | 21.5 | 22.5 | 23.2 | 84.0 | 71.7 | 87.3 | 26.6 | 73.8 | 72.7 | 88.3 | 58.7 |
| Dirichlet | 65.6 | 64.5 | 19.1 | 17.2 | **25.2** | **84.2** | 72.5 | 85.9 | 26.0 | **75.8** | **75.2** | **91.0** | 58.5 |
| Modality dropout | 63.1 | 62.3 | 15.8 | 18.4 | 21.5 | 80.1 | 68.4 | 83.2 | 23.0 | 74.4 | 71.9 | 86.9 | 55.8 |
| OPM | 60.0 | 59.8 | 13.3 | 17.2 | 24.2 | 75.2 | 62.1 | 77.3 | **28.9** | 71.3 | 69.5 | 87.1 | 53.8 |
| MIAM (ours) | **66.0** | 65.0 | **24.0** | **22.7** | 25.0 | 83.4 | **73.8** | **88.3** | 25.2 | **75.8** | 72.1 | 88.3 | **59.1** |
| Oracle (one model per column) | 64.1 | 67.0 | 21.3 | 28.3 | 31.1 | 84.2 | 72.3 | 87.1 | 31.8 | 74.2 | 75.8 | 89.5 | 60.6 |

Table 8: **F1-score on the test set of TaxaBench**. Each column corresponds to a different input subset, showing the performance of each masking strategy on that subset. For each masking strategy, the same trained model is evaluated on all subsets. The best score per subset is written in bold, and the average score across input subsets is reported in the last column.

| Modality | Input Type | | | | | | | | | | | | Avg. |
| --- | --- | --- | --- | --- | --- | --- | --- | --- | --- | --- | --- | --- | --- |
| | Unimodal | | | | | Bimodal | | Trimodal | | Quadri. | | All | |
| Ground-level image | ✔ | | | | | ✔ | ✔ | ✔ | | ✔ | | ✔ | |
| Audio | | ✔ | | | | ✔ | | ✔ | | | ✔ | ✔ | |
| Geographic location | | | ✔ | | | | ✔ | ✔ | ✔ | ✔ | ✔ | ✔ | |
| Environmental features | | | | ✔ | | | | | ✔ | ✔ | ✔ | ✔ | |
| Satellite image | | | | | ✔ | | | | | ✔ | ✔ | ✔ | |
| Uniform | **35.9** | 31.7 | **6.74** | 3.26 | 6.05 | **50.1** | 42.7 | **57.4** | 6.16 | 45.5 | 38.8 | 59.4 | 32.0 |
| Dirichlet | 34.1 | 29.1 | 3.17 | 2.86 | 5.79 | 43.3 | 41.7 | 52.4 | 5.94 | 46.4 | 38.4 | 61.0 | 30.3 |
| Modality dropout | 34.1 | 27.5 | 3.81 | 2.87 | 5.50 | 45.6 | 37.2 | 49.3 | 5.32 | 44.3 | 37.4 | 58.4 | 29.3 |
| OPM | 21.6 | 20.9 | 1.51 | 3.11 | 5.43 | 27.4 | 23.9 | 32.6 | **8.03** | 35.9 | 30.2 | 47.7 | 21.5 |
| MIAM (ours) | 33.7 | **31.9** | 4.13 | **3.92** | **6.50** | 49.1 | **45.1** | 55.4 | 6.93 | **46.9** | **42.0** | **61.2** | **32.2** |
| Oracle (one model per column) | 38.9 | 34.9 | 5.85 | 4.56 | 5.05 | 50.7 | 43.0 | 56.2 | 6.80 | 44.9 | 36.9 | 60.3 | 32.3 |

Table 9: **SatBird dataset results across metrics.** Each column corresponds to a different input subset, showing the performance of each masking strategy on that subset. For each masking strategy, the same trained model is evaluated on all subsets. The best score for each subset is written in bold.

| Modality | MAE [$\times 10e2$] ↓ | | | Avg. | Top-10 acc. ↑ | | | Avg. | Top-30 acc. ↑ | | | Avg. |
| --- | --- | --- | --- | --- | --- | --- | --- | --- | --- | --- | --- | --- |
| Environmental variables | ✔ | | ✔ | | ✔ | | ✔ | | ✔ | | ✔ | |
| Satellite image | | ✔ | ✔ | | | ✔ | ✔ | | | ✔ | ✔ | |
| Dirichlet | **1.93** | 2.26 | 2.13 | 2.11 | **47.2** | 28.5 | 35.9 | 37.2 | **62.9** | 56.0 | 60.0 | 59.6 |
| Modality dropout | 2.02 | 2.26 | **2.05** | 2.11 | 45.4 | **34.5** | **42.9** | **40.9** | 61.4 | 56.7 | 60.8 | 59.6 |
| OPM | 1.97 | **2.21** | 2.11 | **2.09** | 46.3 | 33.5 | 39.1 | 39.7 | 62.2 | 56.2 | 58.9 | 59.1 |
| MIAM (ours) | 1.94 | 2.24 | 2.12 | 2.10 | 45.6 | 32.6 | 40.8 | 39.7 | 62.0 | **57.2** | **61.1** | **60.1** |
| Oracle (one model per column) | 1.91 | 2.20 | 2.03 | 2.05 | 47.8 | 30.6 | 42.0 | 40.1 | 63.2 | 54.8 | 63.4 | 60.5 |

Table 10: **AUC performance of masking strategies on the GeoPlant test set with SSL pre-training under linear probing**. Each column corresponds to a different input subset, showing the performance of each masking strategy on that subset. For each masking strategy, the same trained model is evaluated on all subsets. The best score per subset is written in bold, and the average score across input subsets is reported in the last column.

| Modality | | Partial Unimodal | | | | | Unimodal | | | Bimodal | | | All | Avg. |
|---|---|---|---|---|---|---|---|---|---|---|---|---|---|---|
| Tabular | BIO1 | ✔ | ✔ | | | | ✔ | | | ✔ | ✔ | | ✔ | |
| | WorldClim | | ✔ | | | | ✔ | | | ✔ | ✔ | | ✔ | |
| | Others | | | | | | ✔ | | | ✔ | ✔ | | ✔ | |
| Time series | Clim: 2018 | | | ✔ | ✔ | | | ✔ | | ✔ | | ✔ | ✔ | |
| | Clim: 2000-2018 | | | | ✔ | | | ✔ | | ✔ | | ✔ | ✔ | |
| | Landsat | | | | | | | ✔ | | ✔ | | ✔ | ✔ | |
| Sat. image | Center patch | | | | | ✔ | | | ✔ | | ✔ | ✔ | ✔ | |
| | Others | | | | | | | | ✔ | | ✔ | ✔ | ✔ | |
| Constant | | 70.9 | 80.5 | 81.6 | 82.3 | 65.0 | 82.8 | 84.5 | 68.1 | 84.6 | 76.9 | 80.6 | 81.5 | 78.3 |
| Uniform | | **70.9** | **81.8** | **82.5** | **82.7** | **66.4** | 83.4 | 84.9 | 69.2 | 85.2 | 79.9 | 81.8 | **82.4** | 79.3 |
| Dirichlet | | 68.5 | 82.0 | 78.6 | 82.4 | 59.7 | 83.6 | 83.3 | 64.7 | 83.5 | 80.1 | 78.4 | 79.6 | 77.0 |
| Modality dropout | | 70.9 | 79.2 | 81.1 | 81.9 | 63.9 | 82.5 | 84.8 | 68.7 | 83.9 | 81.2 | 77.4 | 74.9 | 77.5 |
| OPM | | 70.8 | 78.1 | 79.2 | 81.0 | 58.8 | 79.3 | 81.2 | 62.8 | 83.1 | 77.1 | 74.9 | 76.7 | 75.3 |
| MIAM (ours) | | 70.8 | 81.7 | 81.7 | 81.3 | 65.7 | **83.9** | **85.5** | **70.6** | **85.8** | **82.7** | **82.0** | 82.1 | **79.5** |

Unlike GeoPlant, SatBird focuses on 670 bird species and formulates the task as a multi-target regression problem, where the model predicts encounter rates (i.e., values between 0 and 1) at each location. We use the same Summer train/validation/test splits as in Teng et al. (2023). Satellite features are extracted using the same ResNet-18 pretrained on ImageNet (Krizhevsky et al., 2012) as in the original work. For the tabular variables, we use a 3-layer MLP with hidden dimension 256 and ReLU activations. Both networks produce 512-dimensional vectors, which are concatenated and passed through a linear layer followed by a sigmoid to yield the final predictions. The mask token is set to a zero vector. The training procedure follows the code provided by Teng et al. (2023). For MIAM and OPM, modality scores are computed as the inverse of the validation loss.

The results in Table 9 report Mean Absolute Error (MAE) along with top-10 and top-30 accuracy. Overall, no masking strategy consistently outperforms the others: all achieve relatively low MAE, and the best method varies across metrics and modality combinations without a clear trend. As noted earlier, with only two modalities and a single token per modality, masking strategies provide little added value, making this setting less informative for evaluation.

### A.4.4 MIAM FOR SELF-SUPERVISED PRE-TRAINING

MIAM is designed for supervised tasks, but its applicability may extend beyond this setting. In this section, we evaluate the potential of MIAM in a self-supervised pre-training context. The pretext task consists of reconstructing the values of each modality in the GeoPlant dataset, i.e., the tabular variables, time-series entries, and Sentinel-2 image pixels. The transformer encoder and tokenizers follow the same structure as in the main supervised GeoPlant experiment (detailed in Appendix A.1). After encoding, the tokens are passed to modality-specific decoders to reconstruct each modality. Following MultiMAE (Bachmann et al., 2022), but adapted to non-image modalities, we use one decoder per modality. Each decoder consists of layer normalization, a cross-attention layer, another layer normalization, a feedforward network, three transformer blocks, a final linear projection, and a reshape operation to match the format of the target modality. Learned positional embeddings are added to the tokens before entering the decoders to allow the decoder to capture both token position and token type.

This multimodal autoencoder is then trained to reconstruct the modality values using a mean squared error loss, averaged first within each modality and then across modalities. Pre-training is performed on the GeoPlant training set using the given masking strategy that determines which tokens to hide, following the same procedure as in the main supervised experiment (details are in Appendix A.1). We then evaluate the quality of the learned embeddings using linear probing (LP). We follow the

procedure of SMARTIES (Sumbul et al., 2025): a linear classifier is trained for 100 epochs with a learning rate of 0.001 and a batch size of 1024. The linear weights are learned on the GeoPlant validation set using a weighted binary cross-entropy loss optimized with Adam, and performance is then evaluated across the different input subsets.

Since MIAM requires computing a per-modality score to update its modality-imbalance coefficients, we use the reconstruction losses during training as a proxy for modality performance. This choice may not be optimal, but it is straightforward to implement across different setups. We leave the exploration of more principled alternatives to future work.

The results are reported in Table 10. Two masking strategies stand out: the uniform strategy (Zbinden et al., 2026) and MIAM. MIAM performs best when all modalities are present, i.e., in unimodal and bimodal settings, whereas the uniform strategy performs slightly better in partial unimodal setups. Notably, traditional masking strategies for SSL, such as the constant and Dirichlet strategies, underperform, suggesting that they may not be well suited for learning robust representations under missing inputs. These findings indicate that stronger masking strategies tailored to SSL are needed, and that MIAM provides a promising direction.

## A.5 LLM USAGE

We used Large Language Models (LLMs) as assistive tools during the preparation of this manuscript. Their role was limited to polishing grammar, improving clarity, and providing feedback on the coherence of our ideas. They were not involved in generating research questions or designing experiments.

