# OpenReview forum: "MIAM: Modality Imbalance-Aware Masking for Multimodal Ecological Applications"
_ICLR.cc/2026/Conference — ICLR 2026 Poster_

### Official Review · Reviewer_Hfi3 · 2025-10-26

**Soundness:** 3
**Presentation:** 4
**Contribution:** 4
**Rating:** 6
**Confidence:** 3

**Summary:**

The paper introduces a masking strategy called MIAM. The objective is to improve the robustness of transformers to missing modalities at test time. The topic is important because there is a trend towards fusing more and more modalities, while it is crucial to maintain performance on any subset of modalities seen during training for practical utility. MIAM increases the probability of a modality being masked (not given as input) if its unimodal validation performance is high and not changing; conversely, modalities with low and increasing accuracy are masked less often (given as input). MIAM improves over other masking strategies for two multi-modal datasets.

**Strengths:**

- Multi-modal masked modelling is increasingly used, so the topic is important
- I learned from the paper and it may inform my research
- The idea of MIAM is intuitive, fairly simple, and seems to work — at least when the number of modalities > 2
- The paper is well written and has nice figures that aid understanding

**Weaknesses:**

Moderate weakness:
- I believe that according to the appendix, all transformers have 3 layers and embedding dimension 192, which is very very small. For reference, a ViT-Base has 12 layers and dimension 768. I understand that larger models are costlier but I suspect model size will interact with masking strategies. For example, larger models may be able to fit all modalities easier, and thus uniform masking may be fine.

Minor weakness:
- The method seems to rely on computing validation accuracy after each epoch. Thus it cannot be directly used in self-supervised learning, e.g., masked autoencoding (MAE), since we aim to learn task-agnostic representations without using labelled data. Since multi-modal MAE is quite popular, having MIAM directly support MAE would be nice.

**Questions:**

None

---

> ### Author Response · Authors · 2025-11-23
>
> > Moderate weakness: I believe that according to the appendix, all transformers have 3 layers and embedding dimension 192, which is very very small. For reference, a ViT-Base has 12 layers and dimension 768. I understand that larger models are costlier but I suspect model size will interact with masking strategies. For example, larger models may be able to fit all modalities easier, and thus uniform masking may be fine.
>
> Thank you for raising this valid concern. **We have conducted additional experiments varying the model size** by evaluating a relatively large variant (6 layers, 256-dimensional tokens) and a relatively huge variant (12 layers, 512-dimensional tokens) on the GeoPlant dataset. Below, we report the average AUC test performance across all input subsets; the full results, including per-subset scores, are provided in Appendix A.4.1 as **Table 6**.
>
> | Model Size | Constant | Uniform | Dirichlet | Modality Dropout | OPM | MIAM (ours) |
> | --- | --- | --- | --- | --- | --- | --- |
> | **Base** (3L, 192d) | 80.4 | 83.2 | 80.6 | 81.5 | 83.8 | **86.1** |
> | **Large** (6L, 256d) | 78.7 | 82.9 | 81.2 | 82.6 | 81.8 | **85.2** |
> | **Huge** (12L, 512d) | 77.2 | 80.1 | 79.9 | 80.0 | 79.4 | **81.8** |
>
> Across both larger models, **MIAM remains the best masking strategy on average**, outperforming all baselines by a clear margin. However, we also observe a reduction in performance as model size increases, which may indicate overfitting and suggest that additional data would be needed to fully exploit these larger models. **This explanation has been added to Appendix A.4.1 (L966)**.
>
> &nbsp;
>
> > Minor weakness: The method seems to rely on computing validation accuracy after each epoch. Thus it cannot be directly used in self-supervised learning, e.g., masked autoencoding (MAE), since we aim to learn task-agnostic representations without using labelled data. Since multi-modal MAE is quite popular, having MIAM directly support MAE would be nice.
>
> In the paper, we focused our attention on a supervised setup, where a validation set, proxy of the final performance on a test set, was available. To the best of our knowledge, modality imbalance has not been explored in the context of self-supervised learning, so it remains unclear how it would manifest in this setting.
>
> By following the suggestion of the Reviewer, we have **conducted an** **additional experiment to assess the potential of MIAM in a self-supervised learning setup**, where modality performance is estimated using the training reconstruction loss. The experiment is performed on the GeoPlant dataset, using the training set to pre-train a MultiMAE model with MIAM as the masking strategy, and then evaluating with linear probing (LP) on the test set with the LP weights learned on the validation set. Further details are provided in Appendix A.4.4. A summary of the results appears in **Table 4** in the discussion section, with full per-subset results provided in Table 10 in Appendix A.4.4.
>
> Table 4 in the paper: Average AUC performance of masking strategies on GeoPlant with SSL pre-training evaluated via linear probing.
> | **Method** | **Avg.** |
> | --- | --- |
> | Constant | 78.3 |
> | Uniform | 79.3 |
> | Dirichlet | 77.0 |
> | Modality dropout | 77.5 |
> | OPM | 75.3 |
> | MIAM (ours) | **79.5** |
>
> Overall, **MIAM outperforms the other masking strategies under LP, demonstrating its promise for self-supervised settings.** While this remains a small-scale, proof-of-concept experiment, it suggests that exploring multiple masking strategies is important in self-supervised multimodal learning, and that the commonly used Dirichlet distribution may not be optimal, given MIAM’s consistent improvements.

---

### Official Review · Reviewer_CJxa · 2025-10-28

**Soundness:** 2
**Presentation:** 3
**Contribution:** 2
**Rating:** 4
**Confidence:** 4

**Summary:**

The paper introduces MIAM (Modality Imbalance-Aware Masking), a dynamic masking strategy for multimodal learning. MIAM formalizes masking as probability distributions over unit hypercubes and addresses three key principles often missing in prior methods: Full support for all input combinations, Corner prioritization to emphasize critical configurations, and Imbalance awareness by adapting masking probabilities based on modality dominance. To achieve this, MIAM constructs a mixture of product beta distributions and dynamically adjusts masking during training using modality-specific performance and learning speed. This design enables handling arbitrary missing inputs, mitigating modality imbalance, and supporting fine-grained contribution analysis across and within modalities.

**Strengths:**

-	Interesting approach to improve performance and robustness when modalities are missing.
-	Strong average performance across different subsets of modalities compared to other sampling strategies.
-	Methodology is clearly described and well-structured.
-	Provides insightful analysis of how modality-specific values evolve during training.

**Weaknesses:**

-	Only tested on two classification downstream tasks. Broader applicability (e.g., segmentation tasks like flood mapping where optical data is often missing) is not demonstrated. Also includes frequent references to pre-training masking strategies in related work, while not compared for pre-training.
-	Downstream performance is sensitive to new hyperparameters. With ablation shown only on one dataset (Figure 10), it is difficult for readers to tune parameters effectively.
-	Evaluated with only one architecture, which is not well explained, even though MIAM is likely applicable to many others.
-	Training details and architecture description are insufficient. Before presenting results, it should be made explicit that one model per masking strategy was trained and then evaluated on different subsets.

**Questions:**

-	Could MIAM be adapted for other model architectures? If so, what could be challenges?
-	How should practitioners choose λ and κ in practice for new datasets without extensive tuning?

---

> ### Author Response · Authors · 2025-11-23
>
> > Only tested on two classification downstream tasks. Broader applicability (e.g., segmentation tasks like flood mapping where optical data is often missing) is not demonstrated. Also includes frequent references to pre-training masking strategies in related work, while not compared for pre-training.
>
> Across our experiments in the main text and supplementary material, we consider multi-class classification, multi-label classification, and multi-output regression tasks. Exploring broader applicability beyond these settings is an interesting direction for future work. However, by following the Reviewer's suggestion and to illustrate the potential of MIAM as a pre-training masking approach with self-supervised learning, **we have conducted an** **additional experiment to assess the potential of MIAM in a self-supervised learning setup**, where modality performance is estimated using the training reconstruction loss. The experiment is performed on the GeoPlant dataset, using the training set to pre-train a MultiMAE model with MIAM as the masking strategy, and then evaluating with linear probing (LP) on the test set with the LP weights learned on the validation set. Further details are provided in Appendix A.4.4. A summary of the results appears in **Table 4** in the discussion section, with full per-subset results provided in Table 10 in Appendix A.4.4.
>
> Table 4 in the paper: Average AUC performance of masking strategies on GeoPlant with SSL pre-training evaluated via linear probing.
> | **Method** | **Avg.** |
> | --- | --- |
> | Constant | 78.3 |
> | Uniform | 79.3 |
> | Dirichlet | 77.0 |
> | Modality dropout | 77.5 |
> | OPM | 75.3 |
> | MIAM (ours) | **79.5** |
>
> Overall, **MIAM outperforms the other masking strategies under LP, demonstrating its promise for self-supervised settings.** While this remains a small-scale, proof-of-concept experiment, it suggests that exploring multiple masking strategies is important in self-supervised multimodal learning, and that the commonly used Dirichlet distribution may not be optimal, given MIAM’s consistent improvements.
>
> &nbsp;
>
> > Downstream performance is sensitive to new hyperparameters. With ablation shown only on one dataset (Figure 10), it is difficult for readers to tune parameters effectively. **How should practitioners choose λ and κ in practice for new datasets without extensive tuning?**
>
> Thank you for highlighting this important practical point. As described in the text, the two hyperparameters $\lambda$ and $\kappa$ have intuitive interpretations:
> - $\lambda$ controls the tradeoff between dominant and dominated modalities: increasing its value reduces the masking probability of dominated modalities, improving their performance until it eventually reaches a plateau.
> - $\kappa$ determines how strongly the probability mass concentrates near the corners. Small values under-emphasize corner cases, which can hurt setups with few tokens available, since they are seen less often; once $\kappa$ is sufficiently large, its impact diminishes.
>
> While these interpretations are supported by Figure 10, a **new additional experiment on the TaxaBench dataset (Figure 11)** indicates that these hyperparameters are less critical in some settings (notably TaxaBench, which contains only one token per modality), as performance remains stable across different values.
>
> We recommend setting $\kappa = 10$ and choosing $\lambda$ between 1 and 5, depending on the degree of modality imbalance. Note that because $\lambda$ and $\kappa$ parameterize the masking distribution, their effects can be inspected directly by varying their values and visualizing the resulting distributions (e.g., Figure 3 and Figure 9), without requiring model retraining. Naturally, they can also be fine-tuned like any other hyperparameter using a validation set. **We have added a discussion on these recommendations in Appendix A.4.1 (L955).**

---

> ### Author Response · Authors · 2025-11-23
>
> > Evaluated with only one architecture, which is not well explained, even though MIAM is likely applicable to many others. **Could MIAM be adapted for other model architectures? If so, what could be challenges?**
>
> We adopt a standard multimodal architecture in which each modality is first encoded into tokens and then processed by a transformer. **This design is naturally compatible with masking strategies**, since masking simply corresponds to removing or replacing specific tokens before the fusion stage. **We consider two slightly different setups**: for GeoPlant, this corresponds to a mid-fusion setup (multiple tokens per modality), while for TaxaBench it is closer to a late-fusion setup (one token per modality obtained with a encoder).
>
> We use transformers because they have become the *de facto* standard for multimodal learning [1] and for masked modeling approaches such as BERT, MAE, MultiMAE, and 4M. However, MIAM itself is not tied to the transformer architecture: it operates purely at the level of sampling masking probabilities and selecting which tokens to drop. In principle, MIAM can be integrated into any architecture that supports token- or modality-level masking. **In the new self-supervised learning experiment described above, we used an autoencoder setup with an additional decoder architecture**, demonstrating that MIAM has the potential to work well with a variety of model designs.
>
> [1] P. Xu, X. Zhu and D. A. Clifton, "Multimodal Learning With Transformers: A Survey," in *IEEE Transactions on Pattern Analysis and Machine Intelligence*, 2023
>
> &nbsp;
>
> > Training details and architecture description are insufficient. Before presenting results, it should be made explicit that one model per masking strategy was trained and then evaluated on different subsets.
>
> By following the suggestion, **we have revised the text (L362) and captions of Tables 1, 2, 5, 7, 8, and 9** to explicitly state that one model per masking strategy is trained, and that the same trained model is then evaluated across all input subsets (except for the oracle). **We have also expanded the training and architecture descriptions** to ensure all details are present in Appendix A.1. If there is further information you would suggest to add, we are happy to include them. Thanks!

---

### Official Review · Reviewer_5e4q · 2025-11-02

**Soundness:** 2
**Presentation:** 2
**Contribution:** 2
**Rating:** 4
**Confidence:** 4

**Summary:**

This paper introduces MIAM (Modality Imbalance-Aware Masking), a dynamic masking strategy for multimodal learning that addresses the challenge of modality imbalance in ecological applications. The key insight is to formalize masking strategies as probability distributions over unit hypercubes and design a principled approach with three properties: (i) full support over all input combinations, (ii) corner prioritization to favor complete/minimal modality combinations, and (iii) imbalance-awareness that adaptively masks dominant modalities based on their performance and learning dynamics. MIAM uses a mixture of product beta distributions whose parameters are dynamically adjusted during training based on per-modality performance scores and their temporal derivatives. The method is evaluated on two ecological benchmarks (GeoPlant and TaxaBench) and demonstrates consistent improvements over existing masking strategies while providing fine-grained contribution analysis.

**Strengths:**

1. Testing on two diverse ecological datasets (GeoPlant with 3 modalities, TaxaBench with 5 modalities) with multiple modality combinations demonstrates robustness.
2. The fine-grained contribution analysis (Fig. 5) demonstrates how the method can provide ecological insights (e.g., importance of NDVI bands, impact of extreme events), bridging ML and domain science.
3. The progressive ablation showing the contribution of each design principle (uniform hypercube → beta hypercube → MIAM) is convincing.

**Weaknesses:**

1. Only ecological datasets are thoroughly evaluated; broader applicability claims need support from other domains. The SatBird result (Appendix A.4.3) showing similar performance across strategies raises questions about when MIAM is beneficial. No comparison on standard multimodal benchmarks (e.g., vision-language tasks)
2. The choice of ε=3 and φ=10 appears arbitrary. THere is limited discussion of how to set these in practice. The corner weights (Eq. 3) use specific fractions (1/4, 1/2) without justification
3. OPM (Wei et al. 2024) is the main dynamic masking baseline, but other recent modality balancing methods are mentioned but not compared. Missing comparison with recent self-supervised multimodal methods.

**Questions:**

See above

---

> ### Author Response · Authors · 2025-11-23
>
> > Only ecological datasets are thoroughly evaluated; broader applicability claims need support from other domains. The SatBird result (Appendix A.4.3) showing similar performance across strategies raises questions about when MIAM is beneficial. No comparison on standard multimodal benchmarks (e.g., vision-language tasks)
>
> We agree with the Reviewer that our experiments focus on ecological datasets. This choice is intentional: ecological data are particularly challenging and methodologically rich for multimodal learning. These datasets combine several heterogeneous modalities at once — such as interpolated climate variables, geolocation information, long time series, and multi-resolution satellite imagery — which often originate from completely different domains and follow distinct statistical properties. Moreover, ecological applications frequently involve more than two modalities, and often contain multiple tokens per modality (for robustness to missing inputs and for fine-grained contribution analysis). This makes the multimodal learning setting fundamentally different from typical multimodal benchmarks, which are predominantly bimodal (e.g., vision–language) and do not reflect the complexity or diversity of real-world ecological inputs.
>
> In the discussion, we explicitly state that MIAM is most beneficial when:
> 1. **More than two modalities** are available
> 2. **Each modality contains multiple tokens**
> 3. **Modality imbalance** is present.
>
> The SatBird results support this claim: with only two modalities and one token per modality, all masking strategies perform similar, and the task itself has very low error (MAE ≈ 10⁻²). This setup is simply not suitable for evaluating masking strategies, and the experiment confirms that MIAM is not intended for such cases. We include SatBird in the appendix for completeness and to transparently show the regimes in which MIAM is (or is not) effective.
>
> While our empirical evaluation is restricted to ecological datasets — and we do not claim broader applicability beyond the settings studied here — the formalism of masking strategies and the principles introduced by MIAM are general. They have the potential to apply beyond ecology, particularly in domains involving more than two modalities. Exploring such extensions is left for future work, as the scope and diversity of ecological applications already provide a wide and demanding setting.
>
> &nbsp;
>
> > OPM (Wei et al. 2024) is the main dynamic masking baseline, but other recent modality balancing methods are mentioned but not compared. Missing comparison with recent self-supervised multimodal methods.
>
> We indeed mention recent approaches addressing modality imbalance because of their influence on the field. However, although effective at mitigating imbalance, these methods are not masking strategies and therefore **cannot be used to achieve robustness to missing tokens**. They operate in a reduced setting where modality-specific features are linearly combined to produce the final logits, rather than through token-level masking. This is also true for OPM, which we had to adapt for our setup; but unlike the others, OPM defines probabilities to drop entire modalities, allowing comparison with masking-based approaches.
>
> Although our work focuses on supervised learning setups, **we have** **conducted an** **additional experiment to assess the potential of MIAM in a self-supervised learning setup**, where modality performance is estimated using the training reconstruction loss. The experiment is performed on the GeoPlant dataset, using the training set to pre-train a MultiMAE model with MIAM as the masking strategy, and then evaluating with linear probing (LP) on the test set with the LP weights learned on the validation set. Further details are provided in Appendix A.4.4. A summary of the results appears in **Table 4** in the discussion section, with full per-subset results provided in Table 10 in Appendix A.4.4. In particular, **we compare with the masking strategy used in the self-supervised multimodal methods MultiMAE and 4M that rely on the Dirichlet distribution.**
>
> Table 4 in the paper: Average AUC performance of masking strategies on GeoPlant with SSL pre-training evaluated via linear probing.
>
> | **Method** | **Avg.** |
> | --- | --- |
> | Constant | 78.3 |
> | Uniform | 79.3 |
> | Dirichlet | 77.0 |
> | Modality dropout | 77.5 |
> | OPM | 75.3 |
> | MIAM (ours) | **79.5** |
>
> Overall, **MIAM outperforms the other masking strategies under LP, demonstrating its promise for self-supervised settings.** While this remains a small-scale, proof-of-concept experiment, it suggests that exploring multiple masking strategies is important in self-supervised multimodal learning, and that the commonly used Dirichlet distribution may not be optimal, given MIAM’s consistent improvements.

---

> ### Author Response · Authors · 2025-11-23
>
> > The choice of ε=3 and φ=10 appears arbitrary. There is limited discussion of how to set these in practice. The corner weights (Eq. 3) use specific fractions (1/4, 1/2) without justification
>
> We are thankful to the Reviewer for pointing this. As supported by Figure 10, the two hyperparameters $\lambda$ and $\kappa$ have intuitive interpretations:
>
> - $\lambda$ controls the tradeoff between dominant and dominated modalities: increasing its value reduces the masking probability of dominated modalities, improving their performance until it eventually reaches a plateau.
> - $\kappa$ determines how strongly the probability mass concentrates near the corners. Small values under-emphasize corner cases, which can hurt setups with few tokens available, since they are seen less often; once $\kappa$ is sufficiently large, its impact diminishes.
>
> In addition to the results of Figure 10, **we have added a sensitivity analysis of these hyperparameters on the TaxaBench dataset (Figure 11)** which indicates that these hyperparameters are less critical in some settings (notably TaxaBench, which contains only one token per modality), as performance remains stable across different values. We recommend setting $\kappa = 10$ and choosing $\lambda$ between 1 and 5, depending on the degree of modality imbalance. Since $\lambda$ and $\kappa$ parameterize the masking distribution, their effects can be inspected directly by varying them and visualizing the resulting distributions (e.g., Figure 3 and Figure 9), without requiring model retraining. Naturally, they can also be fine-tuned like any other hyperparameter using a validation set. **We added a discussion on these recommendations in Appendix A.4.1 (L955).**
>
> The corners weights $w_c$ follow the principle of allocating half of the total mass to the two key corners, distributing the remainder evenly across the other $2^M - 2$ corners. While we propose specific fractions (i.e., $\frac{1}{4}$ and $\frac{1}{2(2^M - 2)}$), the choice of $w_c$ is flexible: practitioners may prioritize different corners depending on the application (e.g., assigning higher weight to configurations expected to occur more frequently). Nevertheless, to assess the impact of these fractions, we **conducted additional analyses reported in Table 3 of the paper**. Our analysis shows that uniform corner weights lead to lower performance. We also found that reducing the weight of the key corners to $\frac{1}{8}$ for TaxaBench yields a validation accuracy of 36.4, which remains below the performance achieved with our proposed weighting scheme. In principle, this fraction could be treated as a hyperparameter and tuned on the validation set; however, our goal was to define it in a principled and interpretable way.
>
> Table 3 of the paper: Average performance impact of the prioritization of key corners by using a non-uniform $w_c$, evaluated on the validation sets of GeoPlant (AUC) and TaxaBench (Top-1 accuracy).
> |  | **GeoPlant** | **TaxaBench** |
> | --- | --- | --- |
> | Uniform $w_c$ | 85.2 | 36.0 |
> | Non-uniform $w_c$ | **85.4** | **37.1** |

---

### Official Review · Reviewer_Cimh · 2025-11-03

**Soundness:** 4
**Presentation:** 4
**Contribution:** 4
**Rating:** 8
**Confidence:** 5

**Summary:**

This paper proposes a new masking algorithm that addresses the problem of modality competition, wherein one modality dominates the learned features. The algorithm uses a mixture of product beta distributions that concentrations probability mass in the corners (especially the all-on or all-off corners) of a unit hypercube representing the probability that tokens within each modality are masked. The weights are adjusted dynamically to adjust the masking probability depending on the performance and learning speed of each modality. Experiments are performed on two multimodal ecology datasets, GeoPlant and TaxaBench, using a transformer architecture. MIAM overall improves performance on these benchmarks compared to baselines including On-the-fly Prediction Modulation (OPM), which adjusts per-modality probabilities based on relative performance scores but applies the probability to the entire modality, not each token within each modality as in MIAM. The paper also shows how MIAM indicates which inputs drive performance, providing a measure of the contribution/importance of each modality.

**Strengths:**

- This paper addresses an understudied problem in multimodal learning of modality imbalance/competition. Despite being understudied, it is a common challenge in multimodal learning for remote sensing (and other domains).
- The paper is well written and easy to read. Technical details are clearly explained. Figures and tables are high quality and are helpful for understanding the paper’s results and ideas.
- The contribution analysis enabled by MIAM is a nice bonus and would be appreciated by domain experts in ecology and other domains (e.g. across remote sensing applications).
- The ablation experiment is well designed and effectively shows the contribution of each component of the proposed algorithm.
- The proposed algorithm is well-motivated, and the paper gives nice explanations and figures to support the motivation and intuition behind the algorithm’s design.

**Weaknesses:**

- The MIAM masking algorithm could be applied to any model architecture that implements masked multimodal learning. I think there was a missed opportunity to show the value of MIAM on existing multimodal remote sensing foundation models. If it worked, MIAM could significantly improve the utility of these models. (To me, this is the difference between a score of 8 and 10.)
- It seems that a natural extension (or even baseline?) of MIAM is to assign a masking probability to each token, rather than applying the same probability to all tokens within each modality. Did the authors consider or test this?
- I think the motivation for prioritizing the all-on/all-off corners (or corners in general) could be better explained in terms of the ecological application context. It doesn’t seem obvious to me why it would be beneficial to prioritize combinations with almost all tokens or almost no tokens from each modality.

**Questions:**

- Why is it beneficial to prioritize combinations with almost all tokens or almost no tokens from each modality?
- How does MIAM affect the performance of multimodal remote sensing foundation models?
- How does applying the same probability to all tokens in a modality compare to applying a different probability to each token regardless of modality?

---

> ### Author Response · Authors · 2025-11-23
>
> > The MIAM masking algorithm could be applied to any model architecture that implements masked multimodal learning. I think there was a missed opportunity to show the value of MIAM on existing multimodal remote sensing foundation models. If it worked, MIAM could significantly improve the utility of these models. (To me, this is the difference between a score of 8 and 10.) **How does MIAM affect the performance of multimodal remote sensing foundation models?**
>
> Our approach indeed has the potential to be applied to any method that relies on masked multimodal learning, including remote sensing foundation models. For instance, the TerraMind model is based on the 4M framework, which uses the Dirichlet distribution for masking that was outperformed by MIAM in all our experiments. However, while the Beta hypercube variant could directly be plugged into the training of TerraMind, implementing MIAM is not as straightforward, as it requires a performance score for each modality to compute the modality imbalance coefficients. In the supervised setting considered in this paper, these scores can be obtained based on validation performance, but in self-supervised settings, obtaining such scores is not straightforward.
>
> For this reason, **we have** **conducted an** **additional experiment to assess the potential of MIAM in a self-supervised learning setup**, where modality performance is estimated using the training reconstruction loss. The experiment is performed on the GeoPlant dataset, using the training set to pre-train a MultiMAE model with MIAM as the masking strategy, and then evaluating with linear probing (LP) on the test set with the LP weights learned on the validation set. Further details are provided in Appendix A.4.4. A summary of the results appears in **Table 4** in the discussion section, with full per-subset results provided in Table 10 in Appendix A.4.4.
>
> Table 4 in the paper: Average AUC performance of masking strategies on GeoPlant with SSL pre-training evaluated via linear probing.
>
> | **Method** | **Avg.** |
> | --- | --- |
> | Constant | 78.3 |
> | Uniform | 79.3 |
> | Dirichlet | 77.0 |
> | Modality dropout | 77.5 |
> | OPM | 75.3 |
> | MIAM (ours) | **79.5** |
>
> Overall, **MIAM outperforms the other masking strategies under LP, demonstrating its promise for self-supervised settings.** While this remains a small-scale, proof-of-concept experiment, it suggests that exploring multiple masking strategies is important in self-supervised multimodal learning, and that the commonly used Dirichlet distribution may not be optimal, given MIAM’s consistent improvements.
>
> &nbsp;
>
> > It seems that a natural extension (or even baseline?) of MIAM is to assign a masking probability to each token, rather than applying the same probability to all tokens within each modality. Did the authors consider or test this? **How does applying the same probability to all tokens in a modality compare to applying a different probability to each token regardless of modality?**
>
> We agree with the Reviewer that MIAM could indeed be extended naturally to assigning masking probabilities per token. However, we assume that imbalance arises primarily *between* modalities rather than *within* modality; **to the best of our knowledge, no study has shown substantial within-modality imbalance**. That said, the distinction depends on how modalities are defined. For instance, different time series could be treated as separate modalities, in which case MIAM would simply operate over more modalities, each one still represented by a small set of tokens.
>
> We would also like to clarify that extending MIAM to per-token masking probabilities poses a practical limitation: scalability. MIAM estimates modality imbalance coefficients using validation forward passes, and performing this computation for every token **would significantly increase computational cost**.
>
> For these reasons, we focus on the modality level, where MIAM is both well-motivated and tractable. Nonetheless, exploring finer-grained masking is an interesting direction for future work.

---

> ### Author Response · Authors · 2025-11-23
>
> > I think the motivation for prioritizing the all-on/all-off corners (or corners in general) could be better explained in terms of the ecological application context. It doesn’t seem obvious to me why it would be beneficial to prioritize combinations with almost all tokens or almost no tokens from each modality. Why is it beneficial to prioritize combinations with almost all tokens or almost no tokens from each modality?
>
> Thank you for raising this point: it is indeed a key aspect of MIAM and worth clarifying. The motivation for this prioritization stems from both ecological considerations and optimization behavior:
>
> 1. **Prioritizing corners**: In ecological applications, incomplete data often occurs at the level of entire modalities (e.g., missing satellite image of a location due to cloud cover). Prioritizing corners reflects this situation by emphasizing full-presence or full-absence cases. Moreover, tokens from the same modality tend to be highly correlated (e.g., spectral bands, temporal segments), meaning that observing only a few tokens may already capture most of the modality’s information. This can exacerbate modality imbalance if multiple tokens of the dominant modalities are always present, further motivating the need to emphasize complete absence cases.
> 2. **All-tokens (all-on) corner**: It is also common in ecological datasets to have nearly complete observations across all modalities. Ensuring strong performance in these frequent, fully-observed cases is crucial for overall accuracy and generalization. Prioritizing the all-on corner also encourages the model to learn joint cross-modal interactions efficiently when all inputs are available, leveraging complementary information across modalities.
> 3. **Few-tokens (all-off) corner**: Given the high intra-modality correlation, it is important that the model remains capable of extracting useful information even when only a few tokens are available. This is particularly relevant for interpretability: if we wish to assess the contribution of a single token (e.g., a specific tabular variable), the model must have encountered situations in which the corresponding token appeared nearly alone during training.
>
> While these motivations are partly hypothesis-driven, we have also observed empirical support for them. Figure 4 highlights the performance impact on dominated modalities (high difference between Beta and Uniform hypercubes), and **the new ablation study in Table 3 shows consistent performance gains —** especially for dominated modalities like satellite imagery (see Table 5) — when using non-uniform $w_c$. Importantly, as mentioned in the discussion section, **MIAM remains flexible**: for different applications, $w_c$ can be adapted to emphasize alternative corners that better reflect application-specific data patterns.
>
> Table 3 of the paper: Average performance impact of the prioritization of key corners by using a non-uniform $w_c$, evaluated on the validation sets of GeoPlant (AUC) and TaxaBench (Top-1 accuracy).
>
> |  | **GeoPlant** | **TaxaBench** |
> | --- | --- | --- |
> | Uniform $w_c$ | 85.2 | 36.0 |
> | Non-uniform $w_c$ | **85.4** | **37.1** |
>
> We have **clarified this motivation in the main text (L216 and L220)** and **added an extended explanation in the Appendix A.2.2.**

---

### Author Response · Authors · 2025-11-23

**We are thankful to the Reviewers for their thorough and constructive feedback.** We appreciate the recognition of the *importance* of addressing modality imbalance as a *common*, but *understudied challenge* in multimodal learning, especially with masking (Cimh, Hfi3). We are glad that MIAM was found to be *clearly described* (CJxa), *intuitive* (Hfi3), and *well-motivated* (Cimh), and that our ablation studies were perceived as *well-designed* (Cimh), *convincing* (5e4q), and *insightful* (CJxa), helping to explain MIAM’s *strong average performance* across different input subsets compared to other masking strategies (CJxa). We also thank the reviewers for valuing the fine-grained contribution analysis enabled by MIAM, which *demonstrates* *how the method can provide ecological insights* (5e4q) and *would be appreciated by experts in ecology and other domains* (Cimh). Finally, we appreciate the remarks noting that the paper is *well-written*, with *high-quality figures* that *aid understanding* (Cimh, Hfi3).
We are confident that the remaining concerns can be fully addressed through our point-by-point responses during the discussion period. **We have uploaded a revised version of the paper and appendix, with all changes highlighted in blue**. We remain at your disposal if any open questions remain.

---

### Author Response · Authors · 2025-12-02

Dear Reviewers and AC,

We would like to thank you again for the time you dedicated to reviewing our manuscript. We hope that our responses addressed all of your concerns, and we regret that we did not have the opportunity to discuss them further. In particular, the important addition of the new self-supervised learning experiment demonstrating MIAM’s potential in this setting — where it outperforms the baselines, including the commonly used Dirichlet distribution — together with the clarification of the roles of the hyperparameters $\lambda$ and $\kappa$, supported by additional experiments on the TaxaBench dataset, directly addresses the key points raised in the reviews. We believe these revisions strengthen the manuscript and improve its completeness, and for this we are sincerely grateful for your insightful feedback.

---

### Meta-Review · Area_Chair_AFSo · 2026-01-07

**Summary:**

The paper proposes a masking strategy termed Modality Imbalance-Aware Masking (MIAM), which aims to improve the robustness of transformer-based models to missing modalities at test time. MIAM formulates the masking process as a probability distribution over a unit hypercube and is designed to satisfy three key principles that are often overlooked in prior work: full support over all possible input combinations, corner prioritization to emphasize critical modality configurations, and imbalance awareness, which adaptively adjusts masking probabilities according to modality dominance. Experimental results on two multimodal datasets demonstrate that MIAM consistently outperforms existing masking strategies.

The paper received reviews from four reviewers, two of whom were initially inclined toward acceptance. During the rebuttal phase, the authors provided detailed responses, and upon reviewing these rebuttals, I believe that the majority of the reviewers’ concerns have been adequately addressed.

**Reviewer Concerns:**

During the rebuttal phase, the authors addressed the following concerns:
- The applicability of the proposed method and the expansion of the experimental comparisons;
- Hyperparameter selection;
- Model architecture selection.

**Reviewer Scores:**

There was limited discussion among the reviewers; however, the authors addressed most of the reviewers’ comments.

---

### Decision · Program_Chairs · 2026-01-26

Accept (Poster)